# Learning to Extrapolate to New Tasks:
# A Relational Approach to Task Extrapolation

**Adam Ousherovitch** [1]    **Yixin Wang** [1]

## Abstract

Modern learning systems excel at interpolation but struggle to generalize to unseen tasks outside the training distribution's support. This failure occurs even in simple settings, such as handling task parameters beyond the training range, and persists despite advances in foundation models. To this end, we develop the Relational Task Extrapolator (RTE), an algorithm designed to enable systematic extrapolation to novel tasks. The key observation is that extrapolation is inherently relational: extrapolating to unseen tasks requires learning how tasks transform into one another. If a model learns the transformation between tasks A and B during training, it can apply that same transformation to relate known tasks to unseen ones at test time. RTE operationalizes this idea by decomposing each target task into a known anchor task and a transformation linking the anchor and target. It then learns a relational operator, mapping an anchor-transformation pair to predictions for the target task. We instantiate RTE across multiple task extrapolation regimes in function prediction, e.g. where target tasks use out-of-range parameters (parameter extrapolation), have greater compositional depth (length extrapolation), and/or recombine function primitives in unseen ways (compositional extrapolation). We further extend RTE to sequence prediction, integrating it into fine-tuning algorithms for foundation models. Across empirical studies, we find that RTE substantially outperforms existing approaches on extrapolation to novel, unseen tasks.

[1]Department of Statistics, University of Michigan, Ann Arbor. Correspondence to: Adam Ousherovitch <aoushero@umich.edu>.

*Proceedings of the 43rd International Conference on Machine Learning*, Seoul, South Korea. PMLR 306, 2026. Copyright 2026 by the author(s).

## 1. Introduction

Modern learning systems excel at tasks whose data falls within the support of the training distribution. This success has largely been driven by scaling the breadth and diversity of training data, allowing models to interpolate effectively across observed regimes (Kaplan et al., 2020). When test tasks resemble those seen during training, models can exploit statistical regularities to achieve strong performance.

However, when models are asked to generalize to tasks outside the training support, they often fail. Such failures arise even in simple and structured settings, such as extrapolating task parameters beyond the training range, predicting compositions of known functions, or solving tasks generated by longer or more complex processes. Notably, these limitations persist despite advances in large-scale foundation models. Rather than learning the structural rules governing a domain, models instead rely on heuristics tied to the training distribution (Zhang et al., 2021; Malek et al., 2025).

An intelligent system, however, should not require exposure to all possible tasks. If a model internalizes the generative mechanisms underlying a domain, it should be able to reuse those mechanisms to solve new problems (Richens & Everitt, 2024). Yet, even as foundation models improve in scale and capability, they continue to struggle with this form of generalization. For example, a system that can accurately predict planetary trajectories often fails to transfer its implicit understanding of Newtonian mechanics to new physics problems that differ systematically from those seen during training (Vafa et al., 2025).

To this end, we study *task extrapolation*: generalization to target tasks that lie outside the support of the training task distribution. Task extrapolation is distinct from standard notions of out-of-distribution, out-of-support, length, and compositional generalization. In these settings, test inputs are longer or formed by unions of training inputs; in task extrapolation, the task-defining structure itself lies outside the training support. While extrapolating to arbitrary unseen tasks is ill-posed, many real-world tasks are not independent but are systematically related through shared, unknown structure. This motivates our central question: *How can a model apply learned principles to novel tasks*

*whose structure is related to, but not contained within, its training experience?*

**Main idea.** We propose the *Relational Task Extrapolator (RTE)*, an algorithm designed to enable systematic generalization to novel, out-of-support tasks. The key observation is that task extrapolation is fundamentally relational: success depends on learning how tasks transform into one another. If a model learns transformations between tasks during training, it can reuse these learned transformations to relate known tasks to out-of-support ones at test time.

RTE operationalizes this observation via transduction (Vapnik, 1998; Netanyahu et al., 2023). An unseen target task is decomposed into two components that lie within the training distribution: (i) a familiar *anchor task* and (ii) a relative *transformation* linking the anchor to the target. RTE then learns a relational operator to map the anchor, transformation, and query input to predictions for the target task. We illustrate this idea in Figure 1.

Formally, consider a family of functions $f_\theta \in \mathcal{F}$ indexed by task parameters $\theta \in \Theta$. We observe training tasks $y = f_\theta(x), \quad \theta \in \Theta_{\text{train}}$, and aim to predict $y = f_{\theta^*}(x)$ for a target $\theta^*$ outside the convex hull of $\Theta_{\text{train}}$. Rather than directly approximating $f_{\theta^*}$ from limited data, RTE assumes that tasks are related through structured transformations. Specifically, an unseen task $f_{\theta^*}$ can be expressed as a transformation of a known *anchor task* $f_{\theta_{\text{anc}}}$, where $\theta_{\text{anc}} \in \Theta_{\text{train}}$: $f_{\theta^*}(x) = s_{\phi(\theta_{\text{anc}} \to \theta^*)}\big(x, f_{\theta_{\text{anc}}}(x)\big), \forall x$. Here, $s_\phi$ is an unknown transformation operator parameterized by a relative descriptor $\phi$, capturing changes such as parameter shifts, increased compositional depth, or primitives recombination.

Although the transformation operator is unknown, similar transformations exist among training tasks. One training task $f_{\theta_j}$ can often be viewed as a transformed version of another $f_{\theta_i}$, with a corresponding relative change $\phi(\theta_i \to \theta_j)$. RTE exploits this structure by learning a *relational operator* $\Psi$ from pairs of training tasks:

$$f_{\theta^*}(x) = \Psi\left(x,\ f_{\theta_{\text{anc}}},\ \phi(\theta_{\text{anc}} \to \theta^*)\right).$$

By learning how predictions change as tasks vary within the training distribution, $\Psi$ can apply the same transformation rules to extrapolate beyond it.

A practical challenge is selecting the anchor task $\theta_{\text{anc}}$ and the transformation $\phi(\theta_{\text{anc}} \to \theta^*)$. When task descriptors are available, this selection can be straightforward. In many realistic settings, however, neither is observed directly and both must be inferred from sparse observations of the tasks. In Section 2.4, we develop methods that exploit the geometric structure of the task space to infer suitable anchors and transformations.

*Why RTE works.* RTE reframes extrapolation to a fully unseen, out-of-support task as an *out-of-combination* problem.

Both the anchor task and the transformation are individually observed during training, though never composed together in the same way. If a similar transformation exists in the training data, RTE can apply it to a suitable anchor to reach the target task, enabling principled generalization beyond the training support. Fundamentally, this approach relies on the idea that a powerful inductive bias for a model is to understand how it could have gone from one task to another within its training set and replicate that logic at test time.

We emphasize that RTE is not a universal, black-box solution for arbitrary extrapolation. RTE is designed specifically for domains that possess an underlying relational structure, such that we can utilize such structure to extrapolate. As we will show, these are a broad class of problems, but utilizing RTE requires assumptions which we outline in more detail in Section B.

We instantiate RTE across multiple task extrapolation regimes in function prediction, e.g. where target tasks use out-of-range parameters (parameter extrapolation), has greater compositional depth (length extrapolation), and/or recombine function primitives in unseen ways (compositional extrapolation) (Section 3). We further extend RTE to sequence prediction models like large language models (LLMs), integrating it into fine-tuning algorithms (Section 4). Across empirical studies, we find that RTE substantially outperforms existing approaches on task extrapolation. Our implementation and evaluation code are available in our GitHub repository.

**Contributions.** In summary, our main contributions are:

- **Relational Task Extrapolation (RTE):** We propose an algorithm that leverages inter-task relationships to reduce difficult out-of-support extrapolation problems into more tractable out-of-combination ones.

- **Extrapolation Regimes:** We instantiate RTE across three canonical settings: parameter, length, and compositional extrapolation.

- **Training and Inference Protocols:** We develop protocols using Task2Vec-style embeddings and formulate extrapolation as an optimization problem. As a test-time strategy for LLMs, RTE uses the likelihood of sparse context as a proxy reward, enabling search and verification without extrinsic reward signals.

- **Empirical Validation:** We evaluate RTE on synthetic empirical studies and LLM tasks, finding that it consistently improves over existing baselines by exploiting relational structure in data.

**Related work.** This work draws on four themes.

*Transductive Learning.* Our approach, inspired by Netanyahu et al. (2023) with roots in Vapnik (1998), lifts trans-

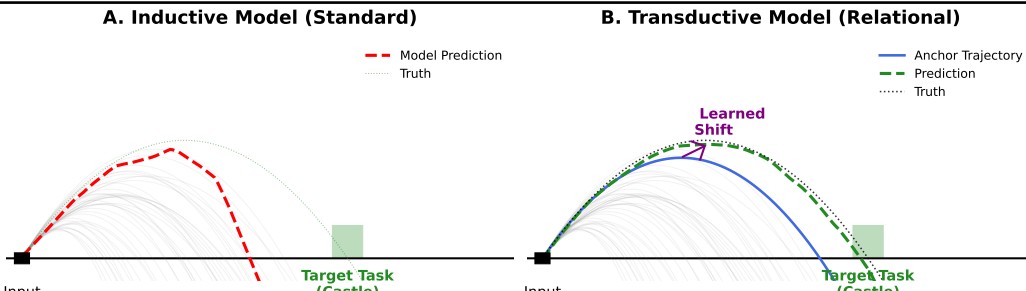

*Figure 1.* **Learning to extrapolate to new tasks via transduction.** Consider a scenario where we have calibrated a cannon by firing a series of "test shots," recording the launch parameters (velocity $v$, angle $\theta$) and the resulting trajectories. The training logs only contain velocities $v \in [30, 60]$. The objective is to hit a target (the "Castle") which requires a velocity of $v = 65$. **(A) Standard Inductive Learning** learns a direct mapping from task parameters to outputs $y = f(x, v, \theta)$ directly, and then applies it independently to new tasks. It saturates at the boundary of its experience and cannot extrapolate, causing the shot to fall short. A model trained on velocities $v \in [30, 60]$ cannot predict the trajectory for a target task $v = 65$. **(B) Transductive Relational Learning (Relational Task Extrapolation (RTE))** predicts for specific test tasks by relating them to known training tasks. It retrieves a known *Anchor Trajectory* (blue, $v = 60$) and learns to apply the relative shift ($\Delta v = +5$, purple arrow) to the curve itself, correctly extrapolating to the Target Task (green), leveraging task-to-task transformations rather than absolute task representations. See Section E.1 for simulation details.

duction from inputs to functions, a much harder scenario since functions lack a natural index, complicating latent relationship inference. Closely related, Hübotter et al. (2024) use transduction for query selection to reduce uncertainty in target regions; RTE instead uses it for *few-shot task inference* without querying new points.

*Statistical Approaches to Extrapolation.* Extrapolation beyond training support is ill-posed without structural assumptions. Pfister & Bühlmann (2024) propose to extrapolate via imposing constraints on directional derivatives at support boundaries; Shen & Meinshausen (2025) enable extrapolation via inverse conditionals in pre-additive noise models; Saengkyongam et al. (2024) show intervention extrapolation is identifiable when latent representations admit linear causal mechanisms. Our work differs from them in mechanism for extrapolation; we posit a *relational* structure: OOD tasks are identifiable through decomposition into known anchors and transformations.

*Generalization and World Models.* Abedsoltan et al. (2025) show Transformers with Chain of Thought generalize by interpolating within shared autoregressive structure, but struggle to *extrapolate* under shifted parameters. Similarly, Zhou et al. (2024) and Ramesh et al. (2023) demonstrate that LLMs fail at length and compositional extrapolation unless forced to align with specific recursive structures (e.g., RASP programs) or perform explicit step-by-step evaluations. RTE is designed to explicitly enforce these structured evaluations algebraically via training mechanisms, without relying on text-based scratchpads at test time. Vafa et al. (2025; 2024) demonstrate that models learn surface heuristics that collapse when physical constants change or when building internal maps. Teoh et al. (2025) address this via Next-Latent Prediction, encouraging consistent world models through latent prediction. Our work differs fundamentally: rather than improving world models via inductive bias, we leverage task-space geometry, projecting known

behaviors into novel regimes via transductive operators.

*Task Embedding and Meta-learning.* Since tasks lack natural indices, we embed them in metric space using the Fisher Information Matrix (Achille et al., 2019). Unlike Ilharco et al. (2023), who apply linear operations on weights, RTE treats inter-task relationships as *learnable*, nonlinear transformations. Meta-learning methods like MAML (Finn et al., 2017) and Reptile (Nichol et al., 2018) learn initializations for fast adaptation but assume new tasks lie within the training distribution's local landscape; RTE targets extrapolation beyond this support.

## 2. Task Extrapolation via Transduction

Standard supervised learning approximates an unknown function $f : \mathcal{X} \rightarrow \mathcal{Y}$ by minimizing risk under a training distribution $\mathcal{D}_{\text{in}}$. This is effective when the test distribution $\mathcal{D}_{\text{test}}$ overlaps the training support. In **out-of-support (OOS)** prediction extrapolation, however, the test support is disjoint from the training support, so inductive predictors typically saturate at the boundary of experience.

Without additional structure, OOS extrapolation is ill-posed: infinitely many hypotheses match $\mathcal{D}_{\text{in}}$ yet disagree arbitrarily on $\mathcal{D}_{\text{out}}$. We explore whether it is possible learn to extrapolate from data by adopting the transductive approach to extrapolation (Netanyahu et al., 2023). Instead of learning a pointwise predictor $\hat{f}(x)$ for each $x$, we learn an *operator* $\Psi$ that predicts $f(x)$ by *anchoring* to an in-support context point $x'$ and applying a *relative transformation* $\Delta x = x - x'$:

$$\hat{f}(x) = \Psi\big(f(x'), \Delta x\big). \qquad (1)$$

When $\Delta x$ is drawn from transformations observed during training, the original OOS query becomes an **out-of-combination (OOC)** instance: both the anchor value $f(x')$ and the transformation $\Delta x$ are in-distribution, even if their

composition is novel. It is important to note that whenever we refer to $\Psi$, we are referring to a fully parameterized model (in our case a neural network) which is trained to minimize loss like Mean Squared Error (MSE) within $\mathcal{D}_{in}$ under the transductive parameterization.

## 2.1. Task extrapolation as transductive prediction

We now lift the transductive principle from *inputs* $x$ to *tasks*, extending prediction extrapolation to task extrapolation. Concretely, we consider a family of functions $\mathcal{F} = \{f_\theta \mid \theta \in \Theta\}$, where $\Theta$ indexes a (possibly latent) task manifold and nearby descriptors induce similar functions. Let $\Theta_{\text{in}} \subset \Theta$ denote the subset covered by the training library. Our goal is to predict under a target task $f_{\theta^*}$ whose descriptor $\theta^*$ lies outside the convex hull of $\Theta_{\text{in}}$. In realistic settings, $\theta^*$ is unobserved (and often $\theta$ is never observed at all); instead, we are given only a sparse (or few-shot) context set $D_{\text{target}} = \{(x_i, y_i)\}_{i=1}^k$ sampled from $f_{\theta^*}$.

Unlike classical domain shift where the relation may be captured by a simple parameter offset $\Delta\theta$, task-to-task relationships can be substantially richer. To make extrapolation tractable, we assume $\mathcal{F}$ is connected by a family of structured transformations $\mathcal{S}$. Each transformation $s_\phi \in \mathcal{S}$ is parameterized by a *relative descriptor* $\phi$, and an unseen target task can be expressed as a transformation of a known *anchor* task $f_{\text{anc}} \in \mathcal{F}_{\text{in}}$:

$$f_{\text{target}} = s_\phi(f_{\text{anc}}). \tag{2}$$

Given few-shot observations from $f_{\theta^*}$, we seek a decomposition $(f_{\text{anc}}, \phi)$ that explains the target context. We then use a learned relational operator $\Psi$ to approximate the action of $s_\phi$, producing predictions for any query input $x$ via

$$\hat{y} = \Psi(x, f_{\text{anc}}, \phi). \tag{3}$$

This reframes task extrapolation as a transductive problem with two components: (i) learning $\Psi$ from transformations observed among training tasks, and (ii) inferring an anchor task and transformation for the target from its sparse context.

## 2.2. Extrapolation Regimes

The relational view of task extrapolation applies to a variety of structural settings, unified by the existence of reusable transformations between tasks. We refer to each such setting as an *extrapolation regime*. Below, we describe three representative regimes that instantiate our model of $\mathcal{F}$; formal definitions and assumptions are provided in Section A.

### 2.2.1. PARAMETER EXTRAPOLATION (CONTINUOUS)

In this regime, task descriptors index a continuous family of functions. Transformations correspond to shifts in descriptor space, with $\phi = \Delta\theta = \theta_{\text{target}} - \theta_{\text{anc}}$. The transformation

operator acts as a difference operator:

$$f_{\theta_{\text{target}}}(x) = s_{\text{diff}}(x, f_{\theta_{\text{anc}}}, \Delta\theta).$$

**Example.** A projectile trajectory under gravity $g = 25$ can be expressed using an anchor trajectory at $g = 20$ together with a shift $\Delta g = +5$. The operator $s_{\text{diff}}$ captures how the trajectory deforms as the gravitational constant increases.

### 2.2.2. LENGTH EXTRAPOLATION (RECURSIVE)

Here, tasks are organized by a notion of hierarchical or recursive complexity $L$. The transformation corresponds to the inductive step that extends a task of complexity $L - 1$ to one of complexity $L$, and $s$ acts as an extension operator:

$$f_{\theta^{(L)}}(x) = s_{\text{ext}}(x, f_{\theta^{(L-1)}}, \phi_L).$$

**Example.** A degree-9 polynomial $P_9(x)$ can be written as its degree-8 truncation $P_8(x)$ plus a new higher-order term $c_9 x^9$. The transformation is the recursive extension step, parameterized by the new coefficient $\phi_L = c_9$.

### 2.2.3. COMPOSITIONAL EXTRAPOLATION (COMBINATORIAL)

In the compositional regime, tasks are formed by combining simpler primitives, e.g., $f_{\text{target}} = f \circ g$. We treat one component as the anchor task and the other as the transformation argument, with $s$ acting as a composition operator:

$$f_{\text{target}}(x) = s_{\text{comp}}(x, f, g).$$

**Example.** The function $h(x) = \sin(x^2)$ is obtained by composing the outer primitive $\sin(\cdot)$ with the anchor primitive $x^2$. Here, $x^2$ serves as the anchor, while the application of $\sin(\cdot)$ defines the transformation.

## 2.3. The Proxy Geometry

Selecting anchors and inferring transformations is straightforward when task descriptors $\theta$ are observed (e.g., gravitational constants or polynomial degrees). In many practical settings, however, $\theta$ is latent: we do not observe task descriptors directly and instead only have access to a dataset $D = \{(x_i, y_i)\}_{i=1}^k$ sampled from the task. To reason about relationships between tasks in this setting, we construct a *proxy geometry*, a task embedding derived solely from data that approximates the structure of the task manifold.

Formally, we define a proxy mapping $\Gamma : \mathcal{D} \to \hat{\Theta} \subset \mathbb{R}^d$, which maps a (possibly few-shot) dataset to a vector representation. For this proxy space to be useful for task extrapolation, $\Gamma$ must satisfy two key properties.

**(i) Topological Fidelity.** The proxy geometry of $\Gamma$ should preserve the intrinsic structure of $\mathcal{F}$. If two tasks $f_1$ and $f_2$ are close in $\Theta$, then their proxy embeddings $\hat{\theta}_1 = \Gamma(D_1)$ and $\hat{\theta}_2 = \Gamma(D_2)$ should be close in $\hat{\Theta}$.

---

**Algorithm 1** Relational Task Extrapolator (RTE)

---

**Require:** Training Library $\mathcal{F}_{\text{train}}$, Target Context $D_{\text{target}} = \{(x_n, y_n)\}_{n=1}^k$, Query $x_{\text{query}}$
**Require:** Parameters: Operator $\Psi$, Proxy Mapper $\Gamma$
**Ensure:** Target Prediction $\hat{y}$
1: *// Phase 1: Learning to Extrapolate*
2: **while** not converged **do**
3:     Sample task pair $(f_i, f_j) \sim \mathcal{F}_{\text{train}}$
4:     Retrieve shift $\phi_{ij}$ {Ground truth or $\Delta\hat{\theta}$}
5:     Sample inputs $x$ from task domain
6:     $\mathcal{L}_{\text{train}} \leftarrow \mathcal{L}(f_j(x), \Psi(x, f_i, \phi_{ij}))$
7:     Update $\Psi \leftarrow \Psi - \eta\nabla\mathcal{L}_{\text{train}}$
8: **end while**

---

9: *// Phase 2: Test Time Decomposition*
10: Compute Target Proxy: $\hat{\theta}_{\text{target}} \leftarrow \Gamma(D_{\text{target}})$
11: **if** Regime is **Continuous then**
12:     $f_{\text{anc}}^* \leftarrow \arg\min_{f \in \mathcal{F}_{\text{train}}} \|\hat{\theta}_{\text{target}} - \hat{\theta}_f\|$
13:     $\phi^* \leftarrow \hat{\theta}_{\text{target}} - \hat{\theta}_{\text{anc}}^*$
14: **else**
15:     Propose candidates $\mathcal{C}_k$ via Decomposer $g_\psi(\hat{\theta}_{\text{target}})$
16:     $(f_{\text{anc}}^*, \phi^*) \leftarrow$
            $\arg\min_{(f,\phi) \in \mathcal{C}_k} \sum_{(x,y) \in D_{\text{target}}} \mathcal{L}(y, \Psi(x, f, \phi))$
17: **end if**
    **return** $\Psi(x_{\text{query}}, f_{\text{anc}}^*, \phi^*)$

---

**(ii) Robustness to Sparsity.** Target tasks are often observed through only a handful of examples (e.g., $k < 10$). Consequently, $\Gamma$ must be stable when estimated from sparse support sets, producing reliable embeddings even in the few-shot regime. In our implementation, we instantiate $\Gamma$ using Task2Vec embeddings derived from the Fisher Information Matrix of a pretrained model (Achille et al., 2019). As discussed in Section C.5, this construction empirically captures the geometry of the task space while remaining sufficiently robust to sparse sampling, provided the underlying model is adequately pretrained.

## 2.4. The Relational Task Extrapolator

We now present the *Relational Task Extrapolator* (RTE), which operationalizes the relational view of task extrapolation introduced above. RTE explicitly separates extrapolation into two stages: a **training phase**, where the model learns how tasks transform into one another within the training library, and an **inference phase**, where a novel target task is decomposed into a familiar anchor task and a transformation consistent with this learned structure.

### 2.4.1. STRUCTURAL ASSUMPTIONS AND LIMITATIONS

We require the following assumptions for RTE.

**1. Combinatorial Reachability.** For extrapolation to be successful, the out-of-domain target task cannot be funda-

mentally alien; it must maintain a relational connection to the training data. Specifically, we assume the target task $f_{\theta^*}$ is reachable: it must be decomposable into an anchor task $f_{\text{anc}}$ and a transformation $\phi$ that both independently exist within the marginal supports of the training distribution, even if their specific combination is novel. In Section B, we discuss how this can be extended to the multi-step case.

**2. Identifiability of Structural Rules.** To effectively learn the relational operator $\Psi$, the mechanism of change must be isolated from the search for the task. Jointly learning both the task manifold and the transformation operator from scratch is an ill-posed problem. Consequently, we assume access to relational metadata during training (e.g., ground-truth pairs indicating $f_j = f_i \circ g$) to ensure the operator converges to the true causal mechanism of transformation rather than memorizing surface heuristics. While this represents a strong requirement, we discuss ways to relax this assumption in Section B. We also present formal guarantees like error bounds in Section C, along with the required assumptions.

### 2.4.2. PHASE 1: LEARNING THE RELATIONAL OPERATOR

In the training phase, we learn a parameterized operator $\Psi$ that approximates the underlying transformation operator $s$. Intuitively, $\Psi$ learns how predictions change as one moves from one task to another. We train $\Psi$ using pairs of tasks sampled from the training library $\mathcal{F}_{\text{train}}$ by minimizing the expected prediction error:

$$\min_\Psi \; \mathbb{E}_{f_i, f_j \sim \mathcal{F}_{\text{train}}} \left[ \mathcal{L}(f_j(x), \Psi(x, f_i, \phi_{ij})) \right],$$

where $\phi_{ij}$ parameterizes the relationship from task $f_i$ to task $f_j$. We consider two settings for specifying $\phi_{ij}$.

**Continuous Regime.** Each training task $f \in \mathcal{F}_{\text{train}}$ is embedded into the proxy space via $\hat{\theta} = \Gamma(f)$. The transformation between two tasks is then defined as a difference in proxy space: $\phi_{ij} \triangleq \Delta\hat{\theta} = \hat{\theta}_j - \hat{\theta}_i$. This reduces relational learning to a transductive problem analogous to the single-function case, where $\Psi$ learns to apply observed shifts in task space.

**Discrete regimes.** In combinatorial regimes (e.g., length or compositional extrapolation), we assume access to ground-truth relational metadata within $\mathcal{F}_{\text{train}}$. For instance, if $f_j = f_i \circ g$, we directly provide the composing function $g$ as the transformation parameter $\phi_{ij}$. We further explore this assumption in Section B.

### 2.4.3. PHASE 2: INFERENCE AS TASK DECOMPOSITION

At test time, we are given a sparse context set $D_{\text{target}}$ sampled from an unseen task $f_{\theta^*}$. Neither the task descriptor $\theta^*$ nor the appropriate anchor task $f_{\text{anc}} \in \mathcal{F}_{\text{train}}$ is known. Our

objective is to find the anchor-transformation pair that best explains the target data under the learned operator $\Psi$.

We formalize this as the following optimization problem:

$$(f_{\text{anc}}, \phi) = \arg\min_{\substack{f \in \mathcal{F}_{\text{train}} \\ \phi \in \Phi_{\text{train}}}} \sum_{(x,y) \in D_{\text{target}}} \mathcal{L}(y, \Psi(x, f, \phi)). \quad (4)$$

Solving (4) directly can be expensive, so we adopt different strategies depending on the structure of the regime. (The complete RTE algorithm is summarized in Algorithm 1.)

**Strategy A: geometric shortcut (continuous).** In the continuous regime, we exploit the proxy geometry to obtain an efficient approximation. We first embed the target context as $\hat{\theta}_{\text{target}} = \Gamma(D_{\text{target}})$. We then: (1) **Select an anchor** by nearest-neighbor search in proxy space: $f^*_{\text{anc}} = \arg\min_{f \in \mathcal{F}_{\text{train}}} \|\hat{\theta}_{\text{target}} - \hat{\theta}_f\|$. (2) **Infer the transformation** as the proxy-space difference: $\phi^* = \hat{\theta}_{\text{target}} - \hat{\theta}^*_{\text{anc}}$. This geometric shortcut converts the decomposition problem into a simple lookup and subtraction in the proxy space.

**Strategy B: amortized discrete search.** In discrete regimes (length and compositional extrapolation), the task manifold is combinatorial, making gradient-based optimization unreliable. Although exhaustive search over $\mathcal{F}_{\text{train}}$ is possible, it does not scale with library size. To address this, we employ *amortized inference*.

Using the known relational structure within $\mathcal{F}_{\text{train}}$ (Section 2.4.2), we train a decomposer network $g_\psi$ that predicts a distribution over plausible anchor-transformation pairs conditioned on the target proxy, i.e., $P(f, \phi \mid \hat{\theta}_{\text{target}})$. At inference time, we restrict the search to the top-$k$ candidates $\mathcal{C}_k$ proposed by $g_\psi$ and solve: $(f^*_{\text{anc}}, \phi^*) = \arg\min_{(f,\phi) \in \mathcal{C}_k} \sum_{(x,y) \in D_{\text{target}}} \mathcal{L}(y, \Psi(x, f, \phi))$. This substantially reduces computation while preserving accuracy; ablation results are provided in Section K.

## 2.5. Sequence Models and Fine-Tuning

We now extend RTE to sequence prediction models, including large language models (LLMs). In this setting, the relational operator $\Psi$ is instantiated as a pretrained sequence model, and task extrapolation is framed as a supervised fine-tuning (SFT) problem that teaches the model to *apply transformations to an anchor context*.

**Relational fine-tuning.** The fine-tuning objective is to train the model to produce the target output $y$ from a prompt that explicitly encodes the relational decomposition. Given an anchor task with demonstrations $D_{\text{anc}}$, a transformation description $\phi$, and a target query $x$, we construct a prompt

$$P = \text{Format}(D_{\text{anc}}, \phi, x). \quad (5)$$

The transformation $\phi$ may be represented as a natural language instruction, a discrete control token, or a learned embedding, depending on the regime. We then fine-tune model

---

**Algorithm 2** RTE Fine-Tuning for Sequence Models

**Require:** Pretrained LLM $M_\theta$, Task Library $\mathcal{T}_{\text{train}}$, Formatting Function $F$
1: **Phase 1: Training (Learning the Operator)**
2: **for** each training step **do**
3:      Sample target task $T_{\text{target}} \in \mathcal{T}_{\text{train}}$
4:      Retrieve ground truth anchor $T_{\text{anc}}$ and shift $\phi$
5:      Sample few-shot examples $D_{\text{anc}} \sim T_{\text{anc}}$
6:      Sample query $(x, y) \sim T_{\text{target}}$
7:      Construct Prompt: $P \leftarrow F(D_{\text{anc}}, \phi, x)$
8:      Compute Loss: $\mathcal{L} \leftarrow -\log M_\theta(y \mid P)$
9:      Update $\theta \leftarrow \theta - \eta\nabla\mathcal{L}$
10: **end for**

11: **Phase 2: Inference (Test Time)**
**Require:** Target Context $D_{\text{target}}$, Query $x_{\text{query}}$
12: *// Solve Eq. 4 via Discrete Search*
13: Initialize candidate set $\mathcal{C}$ from $\mathcal{T}_{\text{train}}$
14: $(T^*_{\text{anc}}, \phi^*) \leftarrow \arg\min_{(T,\phi) \in \mathcal{C}} \sum_{(x',y') \in D_{\text{target}}}$
       $-\log M_\theta(y' \mid F(D_T, \phi, x'))$
15: Sample $D^*_{\text{anc}} \sim T^*_{\text{anc}}$
     **return** $M_\theta(\cdot \mid F(D^*_{\text{anc}}, \phi^*, x_{\text{query}}))$

---

parameters $\theta$ by minimizing the negative log-likelihood of the target sequence:

$$\mathcal{L}_{\text{SFT}} = -\sum_{t=1}^{|y|} \log p_\theta(y_t \mid P, y_{<t}). \quad (6)$$

Training on anchor-target task pairs from the library encourages the model to behave as the relational operator $\Psi$: it learns to reuse the structure implicit in $D_{\text{anc}}$ and apply the transformation $\phi$ to solve the target query $x$.

**Inference and search via likelihood-based decomposition.** At test time, task decomposition poses additional challenges for large sequence models. Computing reliable proxy embeddings for complex sequence tasks is often computationally expensive and unstable in the few-shot regime, even when meaningful geometry exists (see Section J). We therefore adopt a discrete search strategy analogous to Strategy B in Section 2.4.3.

Concretely, we search over candidate anchor–transformation pairs from the training library and score each candidate using the fine-tuned model's likelihood on the target context. This amounts to solving the optimization problem in Eq. (4), with the negative log-likelihood under the model as the loss. The candidate with the lowest loss is selected as the decomposition for the target task.

This procedure can be viewed as a form of test-time adaptation, related in spirit to Chain-of-Thought verification (Wei et al., 2022) and self-consistency or majority voting (Wang et al., 2023), in that predictions are validated against observed support examples. It also bears a high-level resemblance to retrieval-augmented generation (RAG) (Lewis et al., 2020), since we retrieve and condition on auxiliary

*Table 1.* **Parameter Extrapolation Results (MSE on F2).** Inductive methods fail to extrapolate, even with ground truth parameters. Transductive models exploit the manifold geometry to solve the task. ($\pm$ 95% CI).

| FAMILY | T2V INDUCTIVE | INDUCTIVE ORACLE | RTE | TRANSDUCTIVE ORACLE |
|--------|---------------|------------------|-----|---------------------|
| QUADRATIC | $1.20 \times 10^5 \pm 2.47 \times 10^3$ | $1.18 \times 10^5 \pm 2.37 \times 10^3$ | $\mathbf{7.33 \times 10^2 \pm 1.19 \times 10^1}$ | $1.14 \times 10^3 \pm 6.88$ |
| CUBIC | $2.86 \pm 0.12$ | $3.24 \pm 0.13$ | $\mathbf{1.53 \pm 0.09}$ | $0.96 \pm 0.05$ |
| SIN TREND | $0.28 \pm 0.01$ | $0.25 \pm 0.01$ | $\mathbf{0.051 \pm 0.002}$ | $0.055 \pm 0.001$ |
| TRI TREND | $0.46 \pm 0.02$ | $0.48 \pm 0.02$ | $\mathbf{0.048 \pm 0.002}$ | $0.094 \pm 0.003$ |
| EXP | $1.25 \pm 0.10$ | $1.11 \pm 0.09$ | $\mathbf{0.80 \pm 0.07}$ | $1.13 \pm 0.10$ |

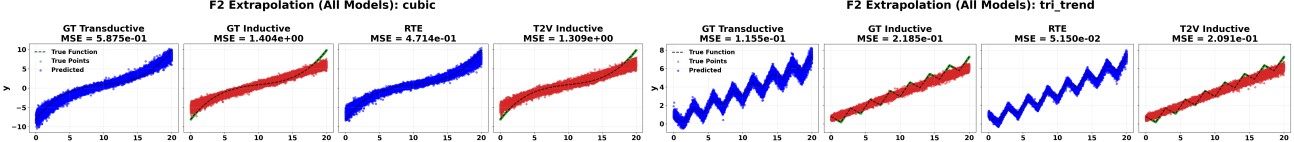

*Figure 2.* **Extrapolation Visualizations. Left:** Cubic. Inductive models (Red) saturate at the boundary of the training support, while transductive models (Blue) capture extreme curvatures at the boundary. **Right:** Tri-Trend. Inductive models fail to capture frequency, while transduction preserves frequency by shifting a known wave anchor. GT denotes Ground Truth or Oracle Parameters

context to minimize prediction loss; accordingly, we sometimes refer to this variant as *Neural RAG*. While our experiments employ brute-force search to isolate the extrapolation mechanism from search efficiency, the same objective can be optimized using more sophisticated procedures such as Bayesian optimization (Snoek et al., 2012) or evolutionary strategies (Salimans et al., 2017).

## 3. Empirical Studies: Synthetic Experiments

We evaluate the Relational Task Extrapolator (RTE) across the three extrapolation regimes introduced in Section 2.2, with the goal of assessing whether RTE can recover out-of-support (OOS) target functions from sparse context more effectively than standard inductive approaches. We further compare RTE against oracle variants that are given access to ground-truth decompositions of the target task, providing an upper bound on achievable extrapolation performance.

Across all synthetic experiments, we instantiate models as standard multi-layer perceptrons (MLPs) (Garg et al., 2022). Concretely, both during training and evaluation, models are required to predict a query output given a small context set of input–output examples from the same task. For RTE, we use Task2Vec embeddings for the proxy geometry. We include detailed experimental setup in Section D.

We find the following: (1) standard inductive baselines fail systematically in all regimes, exhibiting saturation once test tasks move beyond the training support; (2) RTE substantially outperforms inductive models across all settings, confirming that reducing OOS extrapolation to an out-of-combination (OOC) problem alleviates boundary saturation; and (3) although RTE does not match oracle performance, its strong improvement over inductive baselines demonstrates that meaningful task decompositions can be inferred from sparse context alone.

*Table 2.* **Length Extrapolation (MSE on Degree 9).** RTE bridges the gap between the baseline and the Oracle.

| METHOD | MSE |
|--------|-----|
| NAIVE BASELINE | $0.5752 \pm 0.1421$ |
| **RTE** | $\mathbf{0.3712 \pm 0.1021}$ |
| *Oracle* | $0.0497 \pm 0.0093$ |

### 3.1. Parameter Extrapolation

We begin with the continuous parameter extrapolation regime, where task descriptors $\theta$ lie in a region disjoint from the training distribution. We consider five function families (*Quadratic*, *Cubic*, *Exponential*, *Sin-Trend*, and *Tri-Trend*), and partition each parameter space into a training region ($F1$) and a held-out extrapolation region ($F2$).

**Results.** Table 1 highlights a failure mode of inductive learning: *boundary saturation*. On out-of-range parameters, inductive models fail to extrapolate curvature in polynomial tasks (Quadratic, Cubic) and systematically underpredict trends in periodic tasks (Sin-Trend, Tri-Trend). In contrast, RTE bridge this gap by anchoring predictions to known tasks and applying learned transformations. Inductive models also suffer from strong *spectral bias* on periodic functions, failing to recover correct frequencies from sparse context. While larger models or Fourier features (Tancik et al., 2020) can mitigate this issue, RTE bypasses it entirely: RTE inherits the wave structure from the anchor task and only needs to learn a linear shift.

### 3.2. Length Extrapolation

We next evaluate *length extrapolation*, where the objective is to predict degree-$N = 9$ polynomials with $N > N_{\text{train\_max}}$ by extending lower-degree anchors.

**Results.** As shown in Table 2, the Naive Baseline (MSE $\approx$

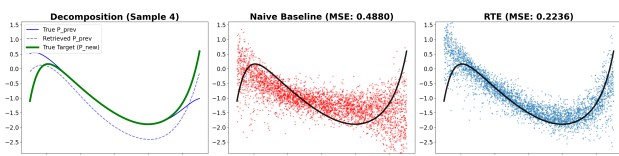

*Figure 3.* **Length Extrapolation (Degree** $8 \to 9$**).** The Naive Baseline (Middle) fails to capture high-curvature dynamics. The RTE (Right) correctly anchors the prediction. The true predecessor and retrieved predecessor are shown (Left)

*Table 3.* **Composition Extrapolation (Aggregate Results).** Comparison of MSE across unseen target pairs.

| METHOD | AGGREGATE MSE |
|---|---|
| NAIVE BASELINE | $0.3891 \pm 0.0378$ |
| **RTE** | $\mathbf{0.2867 \pm 0.0270}$ |
| *Composition Oracle* | $0.1116 \pm 0.0300$ |

0.58) fails to capture the rapid divergence of high-degree polynomials near the boundaries of the input domain. RTE substantially reduces this error (MSE $\approx 0.37$), highlighting the benefit of explicitly modeling the recursive structure underlying length extrapolation. A remaining gap to the Oracle persists, which we attribute to *imperfect retrieval*: the decomposer occasionally selects a suboptimal anchor, limiting the accuracy of the subsequent extension.

### 3.3. Composition Extrapolation

We evaluate RTE on *compositional extrapolation*, testing whether models can generalize to unseen combinations of known primitives (e.g., train on Poly∘Sin; test on Sin∘Poly).

**Results.** Table 3 shows that RTE achieves an aggregate MSE $\approx 0.29$, substantially outperforming the Naive Baseline (MSE $\approx 0.39$). Despite having no access to ground-truth constituent labels, RTE successfully infers functions closely aligned with the underlying primitives.

## 4. Empirical Studies: LLM Experiments

We extend RTE to sequence modeling by instantiating the relational operator $\Psi$ as a fine-tuned instruction-following language model. These experiments examine whether large language models (LLMs) can extrapolate to tasks that require greater logical depth or novel combinations of rules when extrapolation is framed as relational task decomposition rather than direct prediction.

We compare RTE against two baselines: standard fine-tuning, where models are trained to predict targets directly from examples, and oracle variants that are provided with ground-truth logical parents or primitive operations. Across all experiments, we evaluate Qwen (Qwen Team, 2024) and Mistral (Jiang et al., 2023) models fine-tuned using Low-Rank Adaptation (LoRA) (Hu et al., 2022). We consider

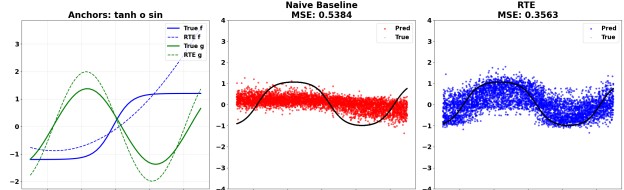

*Figure 4.* **Composition Extrapolation.** The Naive Baseline (Middle) fails to capture the structure of the function. The structure imposed by RTE (Right) finding the pieces of the composition allows for a much more accurate fit. This is even despite RTE not choosing the exact right decomposition (Left)

*Figure 5.* **Transductive Prompting.** Instead of asking the model to solve the target directly, we explicitly provide the Anchor output and the Transformation parameter in the context window. The model learns to apply the logic of the transformation to the anchor.

```
Prompt Strategy for RTE (Sparse Parity)
Instruction:  Apply the Transformation to the Anchor.

-- Anchor Task (Context) --
Input:  [0, 1, 0...]    Anchor Output:   0
Input:  [1, 1, 0...]    Anchor Output:   1

-- Transformation --
Shift Parameter:  <Bit_Index:  3>

-- Target Query --
Input:  [0, 1, 1...]    Target Output:  [PREDICT]
```

two benchmarks: the Sparse Parity task (length extrapolation) from Abedsoltan et al. (2025), and the CodeIO benchmark (compositional extrapolation) from Yuan et al. (2025), where test tasks are structurally distinct from those seen during training. Detailed experimental setup is in Section D.

We find the following: (1) standard fine-tuning struggles to generalize beyond the training regime, failing to track parity over larger input sets and frequently hallucinating incorrect rules for unseen string compositions; (2) RTE substantially improves extrapolation, achieving improved accuracy on Sparse Parity and nearly doubling accuracy on CodeIO relative to the baseline; and (3) RTE remains below the oracle on CodeIO, indicating a residual gap when decomposing complex compositional tasks from sparse supervision.

### 4.1. Sparse Parity (Length Extrapolation)

We evaluate length extrapolation in a logical setting using the Sparse Parity task, which requires computing the parity of a hidden subset of input bits $S$. Models are trained on tasks with subset sizes $|S| \in \{2, \ldots, 5\}$ and evaluated on the out-of-support setting $|S| = 6$.

**Results.** As shown in Table 4, the baseline fails to extrapolate, achieving only $52.86\%$ accuracy and frequently losing track of parity as the subset size increases. In contrast, RTE achieves **66.07% accuracy**. By explicitly decomposing the unseen task into a solved anchor and a learned atomic operation, RTE enables the model to recover and apply the underlying recursive structure.

*Table 4.* **Sparse Parity Extrapolation** ($|S| = 6$). Accuracy on the 28 exhaustive tasks of subset size 6. The Baseline fails to extrapolate to higher complexity, while RTE perfectly recovers the structure.

| METHOD | ACCURACY |
|---|---|
| BASELINE | 52.86% |
| **RTE** | **66.07%** |
| *Upper Bound* | |
| ORACLE (KNOWN PARENT) | 100.0% |

*Table 5.* **CodeIO Results (Exact Match).** Accuracy on 200 unseen compositions. Even with reasoning heuristics, standard inductive LLMs fall short of RTE, which significantly outperforms the baselines.

| METHOD | ACCURACY |
|---|---|
| FEW-SHOT BASELINE | 19.8% |
| COT + MAJORITY VOTE ($N = 16$) | 30.2% |
| **RTE** | **45.3%** |
| ORACLE | 76.0% |

*Figure 6.* **Compositional Prompting.** The prompt defines the retrieved atomic primitives (masked as Func_XX) using the current task inputs. The model must infer that the Target Task is the composition Func_22 ∘ Func_15 (remove vowels from the interlaced string) to solve the query.

```
RTE Prompt Strategy (CodeIO)
You are provided with the following Reference Functions...

-- Func_15 (Primitive A) --
In: banana   Out:  baannaannaab
In: apple    Out:  elppaleppa

-- Func_22 (Primitive B) --
In: banana   Out:  bnn
In: apple    Out:  ppl

Now, use the Reference Functions above to solve the Target Task:
In: banana   Out:  bnnnnnb
In: apple    Out:  lpplpp

### Query:
In: orange   Out:  [PREDICT]
```

### 4.2. CodeIO: Compositional String Transformations

We evaluate *compositional extrapolation* using the CodeIO benchmark, where the model must predict the output of a composite string transformation $T = f_b \circ f_a$ (e.g., applying `sort` followed by `shift`). In this experiment, we also added a Chain of Thought (CoT) (Wei et al., 2022) with Majority Vote (N=16) (Wang et al., 2023) baseline to give RTE stronger competition.

**Results.** The standard Few-Shot Baseline achieves only $19.8\%$ accuracy (Table 5) and the stronger CoT baseline achieved $30.2\%$ accuracy. Qualitative inspection reveals frequent *hallucinations*: the model produces outputs that appear syntactically plausible but follow incorrect transformation rules. In contrast, RTE nearly doubles performance to $45.3\%$. By anchoring generation to known primitives, RTE constrains the model to execute valid algorithmic steps, effectively "reasoning" through the composition.

## 5. Discussion

We introduced a relational approach to task extrapolation. Rather than fitting a single global predictor, RTE learns a local operator over a structured task manifold, enabling neural networks to generalize beyond their training support. Across parameter, length, and compositional extrapolation, we find that RTE can reliably *navigate* unseen regions of the task space when provided with a suitable anchor and a relative direction for transformation.

**Limitations and Future Work.** While RTE reduces intractable out-of-support problems into solvable out-of-combination problems, it introduces practical challenges that could motivate future work:

- **Inference Cost:** Unlike standard inductive models that require a single forward pass, RTE requires test-time compute to identify the optimal decomposition. While we employ amortized search, further improving search efficiency is necessary for scaling to highly complex tasks.

- **Manifold Density (Reachability):** RTE assumes that while the specific target is novel, the *transformation* required to reach it exists somewhere in the training distribution. If a task requires a completely alien mechanism, it remains unreachable.

- **Identifiability and Meta-Labels:** In discrete regimes, RTE currently relies on knowing the ground-truth task relationships during training to isolate the structural rules. While we provide preliminary methods for relaxing this via self-labeling (Section B), inferring latent relational structure directly from scratch remains an open challenge.

- **Multi-Step Extrapolation:** While we demonstrate initial success with deeper compositions and iterative shifts (Section B), scaling the search to reliably chain multiple transformations without compounding errors warrants further investigation.

## Acknowledgments

This work was supported in part by funding from the Office of Naval Research under grant N00014-23-1-2590, the National Science Foundation under grant No. 2310831, No. 2428059, No. 2435696, No. 2440954, a Michigan Institute for Data Science Propelling Original Data Science (PODS) grant, Two Sigma Investments LP, and LG Management Development Institute AI Research. This work used computing resources from the Advanced Cyberinfrastructure Coordination Ecosystem: Services & Support (ACCESS) program, which is supported by U.S. National Science Foundation. Any opinions, findings, and conclusions or recommendations expressed in this material are those of the authors and do not necessarily reflect the views of the sponsors.

## Impact Statement

This paper presents work whose goal is to advance the field of machine learning. There are many potential societal consequences of our work, none of which we feel must be specifically highlighted here.

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

# A. Formal Definitions of Extrapolation Regimes

## A.1. The Relational System

We formally define a **Relational System** as a tuple $\mathfrak{R} = (\Theta, \mathcal{F}, M, \Phi, \mathcal{S})$, where:

- $\Theta$ is a metric space of task descriptors (the parameter manifold).

- $\mathcal{F}$ is the function space, equipped with metric $d_{\mathcal{F}}$.

- $M : \Theta \to \mathcal{F}$ is a generative mapping (the "World Model").

- $\Phi$ is the space of transformation parameters.

- $\mathcal{S}$ is a set of operators $s : \mathcal{F} \times \Phi \to \mathcal{F}$.

**Assumption A.1 (Structural Regularity).** For the system to support extrapolation, we assume the mapping $M$ satisfies *Metric Regularity*. There exists a constant $L < \infty$ such that for all $\theta_1, \theta_2 \in \Theta$:

$$d_{\mathcal{F}}(M(\theta_1), M(\theta_2)) \leq L \cdot d_{\Theta}(\theta_1, \theta_2). \tag{7}$$

This ensures that local neighborhoods in the parameter manifold $\Theta$ map to local neighborhoods in the function space $\mathcal{F}$.

## A.2. Reachability and Extrapolation

Let $\Theta_{\text{train}} \subset \Theta$ be the set of descriptors observed during training. We define the **Training Support** as the $\epsilon$-cover of the image of $\Theta_{\text{train}}$ in $\mathcal{F}$. A target task $f^* = M(\theta^*)$ is considered **Out-of-Support** if $\min_{\theta \in \Theta_{\text{train}}} d_{\Theta}(\theta^*, \theta) > \epsilon$.

**Definition A.1 (1-Step Transductive Reachability)** *An Out-of-Support target task $f^*$ is **1-Step Reachable** if there exists a decomposition $(f_{anc}, s, \phi)$ such that:*

$$f^* = s(f_{anc}, \phi) \tag{8}$$

*satisfying:*

1. ***Anchoring:*** *$f_{anc} \in M(\Theta_{train})$.*

2. ***Transformation Consistency:*** *The parameter $\phi \in \Phi$ represents a valid transition in the training distribution. Formally, $\phi$ must belong to the support of transformations observed in training: $\phi \in sup(P(\Phi|\mathcal{D}_{train}))$.*

## A.3. Regime Instantiations

We instantiate the abstract spaces for our three experimental regimes.

### A.3.1. REGIME I: PARAMETER EXTRAPOLATION

- **Structure:** $\Theta$ is a vector space $\mathbb{R}^d$.

- **Transformation ($\Phi$):** The difference vector $\delta \in \mathbb{R}^d$.

- **Operator:** $s(f, \delta) = M(M^{-1}(f) + \delta)$.

- **Reachability:** A target is reachable if it lies on a translation of a known anchor, $\theta^* = \theta_{\text{anc}} + \delta$.

### A.3.2. REGIME II: LENGTH EXTRAPOLATION

- **Structure:** $\Theta$ is a graded set $\bigcup_{L=1}^{\infty} \Theta^{(L)}$.

- **Transformation ($\Phi$):** The innovation parameter $c$ required to extend complexity from $L$ to $L+1$.

- **Operator:** The recursive step $s_{ext} : \mathcal{F}^{(L)} \times \Phi \to \mathcal{F}^{(L+1)}$.

- **Reachability:** $f^* \in \mathcal{F}^{(L)}$ is reachable if it decomposes into a known predecessor $f_{\text{anc}} \in \mathcal{D}_{\text{train}}$ such that $f_{\text{anc}} \in \mathcal{F}^{(L-1)}$ and valid innovation $\phi \in \Phi$

### A.3.3. REGIME III: COMPOSITIONAL EXTRAPOLATION

- **Structure:** $\mathcal{F}$ is closed under a binary operation $\circ$.

- **Transformation ($\Phi$):** The space of functions itself, $\Phi \equiv \mathcal{F}$.

- **Operator:** The composition $s(f, g) = f \circ g$.

- **Reachability:** A target is reachable if it factors into constituents $f, g$ that are individually present in $\mathcal{D}_{\text{train}}$.

## B. Minimizing Assumptions

A primary bottleneck of using RTE lies in its structural prerequisites: the assumption of single-step combinatorial reachability and the reliance on ground-truth relational metadata (meta-labels) during the training phase. To broaden the applicability of RTE, we outline mechanisms and preliminary experiments demonstrating how these assumptions can be systematically relaxed.

### B.1. Multi Step Extrapolation

While our primary experiments focus on single-step extrapolation, target tasks in real-world settings may require multiple intermediate transformations that are not individually captured by a single anchor-transformation pair. For instance, in parameter extrapolation, navigating to a distant target may require applying a shift vector $\Delta\theta$ iteratively to remain within the local bounds of the transformation operator. Similarly, length and compositional extrapolation may require chaining multiple recursive steps or primitive operations.

### B.1.1. PARAMETER EXTRAP

First, we provide an experiment to show how performance maintains in the parameter extrapolation setting as we move further from the training domain. We extended the continuous function families from Section 3.1 by defining progressively distant out-of-support regions: $F3$, $F4$, and $F5$. To reach these distant targets, RTE must apply its learned 1-step operator iteratively.

Because the intermediate functions are purely latent and have no ground-truth context to condition the model's forward pass, we utilize a "ghost context" interpolation strategy. We linearly interpolate the latent shift vector $\Delta\hat{\theta}$ into $N$ safe steps. Correspondingly, we hallucinate the sparse target context at step $t$ by linearly interpolating between the initial anchor's observations and the final distant target's observations. The model predicts the intermediate curve, which is then stitched together and passed as the anchor for the subsequent step $t + 1$. This ultimately means that we are applying the full operation $\Delta\hat{\theta}$ in N pieces, which are all in distribution transformations.

We evaluate a 3-step extrapolation to region $F4$ and a 4-step extrapolation to region $F5$, comparing RTE against the standard Sparse Inductive Baseline.

*Table 6.* **Multi-Step Parameter Extrapolation (MSE).** Comparison of compounding error over multiple iterative steps into distant out-of-support regions. RTE degrades gracefully compared to inductive saturation.

| FAMILY | METHOD | F4 (3-STEP) | F5 (4-STEP) |
|---|---|---|---|
| QUADRATIC | INDUCTIVE BASELINE | $4.52 \times 10^5$ | $1.09 \times 10^6$ |
| | **RTE** | $\mathbf{1.63 \times 10^5}$ | $\mathbf{4.94 \times 10^5}$ |
| CUBIC | INDUCTIVE BASELINE | 25.72 | 108.63 |
| | **RTE** | **4.83** | **32.26** |
| EXP | INDUCTIVE BASELINE | 85.48 | 463.80 |
| | **RTE** | **35.45** | **391.17** |
| TRI-TREND | INDUCTIVE BASELINE | 9.69 | 29.22 |
| | **RTE** | **2.05** | **7.33** |
| SIN-TREND | INDUCTIVE BASELINE | 10.28 | 21.35 |
| | **RTE** | **5.75** | **14.91** |

**Results.** As expected, compounding error affects both models as the target distance increases ($F5 > F4$). However, the standard inductive models, having saturated at the boundary of the training support, completely explode when asked to predict these distant domains. In contrast, by navigating the manifold iteratively, RTE maintains a meaningful structural signal, consistently achieving between one-half to one-fifth of the error of the inductive baseline across all function families.

### B.1.2. COMPOSITIONAL EXTRAPOLATION

Next, we evaluate multi-step compositional extrapolation, where the target task is a deeper composition of primitives (e.g., depth $d = 3$, such that $f^{(3)} = h \circ g \circ f$). In this discrete setting, the burden of added complexity falls entirely on the decomposition search.

**Experimental Setup**    To test systematic compositional generalization, we construct a task library using three underlying function families: polynomials, sinusoids, and hyperbolic tangents. We enforce a strict out-of-distribution (OOD) holdout strategy during training: the model is never exposed to any sequence where a hyperbolic tangent is applied directly to a sinusoid. Out of the 27 possible depth-3 combinations, this leaves 6 strictly unseen compositional recipes reserved entirely for test-time evaluation. This is very similar to our setup for Section 3.3.

**Multi-Step Decomposition Search**    To navigate this multi-step combinatorial space without suffering from compounding autoregressive errors, our Relational Task Extrapolator (RTE) employs a parallelized decomposition and verification strategy at inference time:

1. **Parallel Latent Prediction:** A *Parallel Decomposer* network processes the sparse target context. Instead of predicting the composition step-by-step, it uses a feed-forward architecture to simultaneously predict a sequence of latent embeddings $z_1, z_2, \ldots, z_d$, representing the primitive functions at each depth of the composition. It is also trained on the identity function to allow it to learn a latent embedding it can use to pad the depth.

2. **Candidate Retrieval:** For each depth layer, the system retrieves the top-$k$ nearest neighbor candidate functions from the training library based on Euclidean distance in the proxy Task2Vec embedding space.

3. **Test-Time Verification (Re-ranking):** The system performs a combinatorial search over the retrieved candidate sequences. For each potential sequence of primitives, it iteratively computes the composite output on the sparse context points. This is done using linear interpolation over the retrieved discrete curve data, meaning we never need to know the underlying analytic functions. The candidate sequence that minimizes the MSE against the provided sparse context is selected as the final predicted decomposition.

**Results**    We evaluate RTE against a standard Inductive baseline over 1,000 trials, evaluating strictly on the held-out OOD target tasks.

| Model | Mean MSE | 95% CI | Std Dev |
|---|---|---|---|
| Inductive (Base) | 16.0375 | $\pm$ 0.4571 | 7.3752 |
| RTE (Recursive) | 12.5424 | $\pm$ 0.6078 | 9.8067 |

*Table 7.* Multi-step compositional extrapolation ($d = 3$) evaluated on 6 systematically held-out OOD recipes across 1,000 trials.

The Inductive baseline struggles to generalize to the novel sequential structures, exhibiting a high MSE of 16.0375. Because it attempts to learn a direct mapping from the sparse context to the target outputs, it saturates when encountering structural recipes absent from its training support.

In contrast, RTE significantly reduces this error to an MSE of 12.5424. By relying on its parallel decomposer to search the proxy geometry and explicitly test combinations of familiar primitives against the context points, RTE successfully navigates the deeper compositional gap without requiring full end-to-end exposure.

### B.2. Meta Labels for Discrete Regimes

The assumption of fully supervised ground-truth relationships during training guarantees the identifiability of the relational operator, but it limits applicability when task generation mechanics are entirely latent. To relax this requirement, we explore

whether the model can infer its own meta-labels using an Expectation-Maximization (EM) inspired training procedure.

If we treat the underlying relationship $\phi$ as latent during training, the learning objective becomes a joint optimization problem:

$$\min_{\Psi,\phi} \mathcal{L}(f(x), \Psi(x, f, \phi)) \tag{9}$$

where $f$ is the target function we wish to predict. This formulation naturally lends itself to an EM-style approach.

Taking compositional extrapolation as an example, this EM procedure alternates between two steps:

1. **E-Step (Self-Labeling):** The model receives a random composite target task from the training set. Lacking the ground-truth decomposition, it scores candidate decompositions (excluding the trivial identity decomposition) to identify which combination minimizes the prediction loss on the provided support examples. The lowest-loss decomposition is assigned as a pseudo-label. Intuitively, the decomposition that makes the target easiest to predict is deemed the correct structural label.

2. **M-Step (Relational Fine-Tuning):** The model parameters are updated via supervised fine-tuning, treating the self-generated pseudo-labels as the ground-truth structural relationships.

A key bottleneck in this approach is the E-Step: identifying accurate pseudo-labels from scratch is nearly impossible unless the model already possesses some foundational understanding of how to apply transformations. To overcome this cold-start problem, we introduce two warm-start strategies:

**1. Primitive Warm-up:** Prior to the EM loop, the model is briefly fine-tuned exclusively on the primitive, atomic transformations. This initializes the relational operator with a working knowledge of the basic building blocks, giving it sufficient signal to score combinations meaningfully during the first E-Step.

**2. Semi-Supervised Relational Labels:** In scenarios where primitives are unavailable, we provide the model with a small subset of the ground-truth relationship labels (e.g., 25% of the training data). This anchors the model's understanding of the relational structure, allowing it to reconstruct the remaining missing labels during the EM process.

**Experiments and Results.** We evaluated these relaxed-assumption procedures on the CodeIO compositional extrapolation benchmark. Despite removing full supervision of the relational structure, both EM-based approaches recover the majority of the performance gap between standard inductive baselines and the fully supervised RTE framework.

Using the Primitive Warm-up EM strategy, RTE achieved an accuracy of 40.625%, significantly outperforming the Chain-of-Thought with Majority Voting baseline (30.21%). Similarly, the Semi-Supervised strategy (utilizing only 25% of the ground-truth labels) successfully reconstructed the missing task graph to achieve 41.67% accuracy.

Ultimately, relaxing these structural assumptions incurs a significant computational cost, but it highlights an important theoretical property of the framework. Taken to its logical extreme, even if a system completely lacks access to both primitives and meta-labels, the relational structure could theoretically be bootstrapped entirely from scratch. Given a sparse target context, one could exhaustively train models across all possible relational assignments and select the configuration that minimizes prediction error on the test set. While such a combinatorial search is computationally intractable for large task libraries, it demonstrates that task extrapolation without structural priors reduces fundamentally to an optimization problem over relational labels.

## C. Theoretical Guarantees for Relational Task Extrapolation

In this section, we provide the theoretical justification for why the Relational Task Extrapolator (RTE) can generalize to Out-of-Support tasks.

**The Intuition:** Standard generalization bounds assume test data comes from the same distribution as training data. Since our test tasks are disjoint from the training set, these bounds do not apply. Instead, we rely on a **Transductive Bound** following the work of Netanyahu et al. (2023). They show if a function has a low-rank structure (i.e., values of the function are formed by predictable combinations of anchors and transformations), learning a transductive operator $\Psi$ is equivalent to Matrix Completion. Thus, even if a specific target value (combination) is unseen, if its constituent parts (anchor and transformation) have been observed in other contexts, the error is bounded. We take their bound and lift it to the task space to show an error bound for RTE for the parameter extrapolation regime.

## C.1. Problem Formulation

Let $\mathcal{F}$ be a family of functions indexed by a parameter space $\Theta$. A specific task is defined by $f_\theta \in \mathcal{F}$.

- **Ground Truth:** We assume there exists a ground truth evaluation operator $H_*$ that maps a task descriptor $\theta$ and input $x$ to an output:
$$H_*(\theta, x) = f_\theta(x).$$

- **Relational Decomposition:** We do not observe $\theta$ directly at test time. Instead, we posit that a target task $\theta_{\text{target}}$ can be reached from a known anchor $\theta_{\text{anc}}$ via a transformation $\phi$, such that $\theta_{\text{target}} \approx \theta_{\text{anc}} + \phi$.

- **Learned Operator:** Our model, RTE, learns an operator $\Psi$ that predicts the output based on the anchor function and the transformation parameter:
$$\hat{y} = \Psi(x, f_{\theta_{\text{anc}}}, \phi).$$

Our goal is to bound the risk (expected squared error) on a target distribution $\mathcal{D}_{\text{target}}$ consisting of tasks never seen during training.

## C.2. Structural Assumptions

To guarantee extrapolation, the task manifold must satisfy three properties: Smoothness, Coverage, and Simplicity.

**Assumption 1: Manifold Smoothness (Lipschitz Continuity).** The mapping from task parameters to function outputs must be smooth. Small errors in identifying the task geometry should not result in catastrophic errors in prediction. Formally, let $d_\Theta$ be a metric on the parameter space. We assume the evaluation $H_*$ is $L_M$-Lipschitz continuous with respect to $\theta$:

$$|H_*(\theta_1, x) - H_*(\theta_2, x)| \leq L_M \cdot d_\Theta(\theta_1, \theta_2) \quad \forall x. \tag{10}$$

**Assumption 2: Combinatorial Coverage.** We cannot extrapolate to a task that is completely alien. We assume the target tasks are *new combinations* of *known components*. Let the training set support marginal distributions over anchors ($\mathcal{D}_{\text{anc}}$) and transformations ($\mathcal{D}_\phi$). We assume the target distribution $\mathcal{D}_{\text{target}}$ is covered by the product of these marginals:

$$\mathcal{D}_{\text{target}} \ll_\kappa \mathcal{D}_{\text{anc}} \otimes \mathcal{D}_\phi. \tag{11}$$

*Interpretation:* If we have seen Anchor A (paired with Transform 1) and Transform 2 (paired with Anchor B), we can extrapolate to the unseen combination of Anchor A + Transform 2.

**Assumption 3: Low-Rank Task Interaction.** The mechanism of how tasks change must be simple. We assume the ground truth operator $H_*$ is *bilinearly transducible*. This means the interaction between an anchor and a transformation can be approximated by a low-rank factorization of dimension $r$:

$$H_*(\theta_{\text{anc}} + \phi, x) \approx \langle u(\theta_{\text{anc}}), v(\phi) \rangle_x. \tag{12}$$

*Interpretation:* This assumption turns the problem into Matrix Completion. If we view tasks as entries in a matrix (Rows=Anchors, Cols=Transformations), this assumption states the matrix is low-rank, allowing us to fill in missing entries (unseen tasks) based on observed entries.

## C.3. Extrapolation Bound

Under these assumptions, we adapt the matrix completion bounds from Netanyahu et al. (2023) to the functional setting.

**Theorem C.1 (Relational Extrapolation Bound)** *Let $\mathcal{R}_{train}$ be the risk achieved by the operator $\Psi$ on the training set. The risk on the unseen target distribution $\mathcal{D}_{target}$ is bounded by:*

$$\mathcal{R}_{target} \leq \underbrace{L_M^2}_{Smoothness} \cdot \underbrace{\kappa^2 \left(1 + C \frac{M_*^4}{\sigma_*^4}\right)}_{Combinatorial\ Factor} \cdot \mathcal{R}_{train}. \tag{13}$$

**Analysis of Terms:**

1. **Training Risk ($\mathcal{R}_{\mathbf{train}}$):** The base performance of the model on seen pairs.

2. **Smoothness ($L_M^2$):** Measures how volatile the function family is. If task structure changes quickly as a function of $\theta$, prediction is difficult.

3. **Combinatorial Factor:** This term coming from Netanyahu et al. (2023) dominates the bound. It depends on $\kappa$ (how "new" the combinations are) and the ratio $M_*/\sigma_*$ (the signal-to-noise ratio of the latent low-rank structure).

## C.4. Applicability to Discrete Regimes

Does this bound apply to Length and Compositional Extrapolation? **Yes.** The theorem relies on the complexity of the interaction between the anchor and the transformation. So long as that interaction is low rank, the "matrix completion" proof of Netanyahu et al. (2023) still holds.

As a couple examples:

1. **Length Extrapolation:** Consider the space of polynomials. The "transformation" $\phi$ is the addition of a high-order term $c_n x^n$.

   - *Metric Regularity:* Polynomial evaluation is Lipschitz continuous with respect to coefficients on bounded domains.
   - *Reachability:* If the training set contains examples of degree extension (e.g., $P_2 \to P_3$, $P_5 \to P_6$), then the "Extension Operator" is in-support.
   - *Low Rank:* The operation is additive (linear), which is rank-1. Thus, the bound holds.

2. **Compositional Extrapolation:** Consider $h(x) = f(g(x))$.

   - *Reachability:* If $f$ and $g$ appear in the training set (paired with other functions), the combinatorial coverage condition ($\mathcal{D}_{\mathrm{target}} \ll_\kappa \mathcal{D}_{\mathrm{anc}} \otimes \mathcal{D}_\phi$) is satisfied.
   - *Low Rank:* Composition is not linear, but often admits low-rank structure in embedding spaces (e.g., in reproducing kernel Hilbert spaces). If the operator $\Psi$ has sufficient capacity to linearize the interaction (Assumption 3), the bound applies.

Therefore, RTE is theoretically justified for any regime where the target task can be decomposed into components that are *individually* familiar, provided the mapping from task descriptors to outputs is stable.

## C.5. Conditions on the Proxy and Justification for Task2Vec

The bound in this section requires *Metric Regularity*. Therefore, our proxy embeddings, $\Gamma(D_f) = \hat{\theta}$, ideally should have this property as well

Specifically, the proxy mapping $\Gamma$ should satisfy a **Metric Isometry Condition** globally with respect to the function family $\mathcal{F}$. Let $d_{\mathcal{F}}(f_i, f_j)$ be a distance measure on the function outputs. The proxy embedding is valid for extrapolation if there exists a constant $C > 0$ such that for any two tasks $i, j$:

$$d_{\mathcal{F}}(f_i, f_j) \leq C|\hat{\theta}_i - \hat{\theta}_j|. \tag{14}$$

**Task2Vec as a Metric-Preserving Proxy.** We utilize the Task2Vec embedding (Achille et al., 2019) as our choice for $\Gamma$. While Task2Vec does not provide a strict global isometry, meaning the embedding distance does not perfectly match semantic distance globally, it does have similar properties locally.

First, the Fisher Information Matrix (FIM), from which the embedding is derived, is a Riemannian metric on the space of probability distributions. It locally quantifies the information a parameter contains about the joint distribution of the task. Because the embedding is derived from the FIM, it serves as an upper bound to the Hessian of the cross-entropy loss. This ensures that the embedding norm scales with task complexity, preserving the local geometry of the loss landscape.

Second, while strict isometry is not guaranteed, empirical evidence demonstrates that the cosine distance between Task2Vec embeddings correlates positively with natural semantic metrics, such as taxonomic distance in biological classification. The embedding successfully recovers hierarchical clusters even across disjoint domains. This suggests that while $\Gamma$ may not be a perfect linear map, it preserves the topological neighborhood structure required for the relational operator $\Psi$ to identify valid anchors, which can be seen in Section H and Section J.

## D. Experimental Setup in Empirical Studies

### D.1. Parameter Extrapolation (Section 3.1)

**Baselines and methods.** We compare the following approaches: (1) **T2V Inductive**, which conditions on the sparse context together with the inferred Task2Vec embedding $\hat{\theta}$, testing whether a task signature alone suffices for extrapolation; (2) **Inductive Oracle**, which is given the sparse context and the *ground-truth* target parameter $\theta_{\text{target}}$, testing whether the model can predict outside the training range; (3) **RTE**, which uses the inferred proxy embedding $\hat{\theta}$ to retrieve an anchor task from the training library and compute a projected shift $\Delta_{\text{proj}}$; and (4) **Transductive Oracle**, which is provided with the correct anchor task $f_{\text{anc}}$ and the exact parameter difference $\Delta\theta$.

### D.2. Length Extrapolation (Section 3.2)

**Baselines and methods.** We compare three approaches: (1) a **Naive Baseline**, which uses a standard MLP to predict the target outputs directly from the sparse context; (2) **RTE**, which employs a Decomposer to retrieve candidate degree-$N-1$ anchor polynomials from the training set, tentatively applies the extension step, and verifies consistency with the observed context; and (3) an **Oracle**, which is provided with the true anchor polynomial $P_{N-1}$ and the exact new coefficient $c_N$.

### D.3. Composition Extrapolation (Section 3.3)

**Baselines and methods.** We compare three approaches: (1) a **Naive Baseline**, which directly regresses the target outputs from the sparse context; (2) **RTE**, which the target $h(x)$ into latent embeddings $z_f, z_g$, retrieves nearest neighbors from the training library, and composes them; and (3) a **Composition Oracle**, which is given access to the exact constituent values $f(x)$ and $g(x)$ for each query.

### D.4. Sparse Parity (Length Extrapolation) (Section 4.1)

**Baseline and methods.** We view a size-6 parity task as a recursive extension of a size-5 task: $f_{S\cup\{k\}}(x) = f_S(x) \oplus x_k$. A Qwen-2.5-7B model is fine-tuned with LoRA to act as the relational operator $\Psi$. We compare three settings: (1) a **Baseline**, which predicts outputs directly from examples; (2) **RTE**, which at test time searches over anchor–bit pairs from the training library and selects the pair minimizing loss on the few-shot context; and (3) an **Oracle**, which is explicitly provided with the correct parent task and the index of the additional bit.

### D.5. CodeIO: Compositional String Transformations (Section 4.2)

**Experimental setup.** We construct a library of 33 atomic string primitives and generate all unique ordered pairs of primitives ($|\mathcal{A}| \times |\mathcal{A}|$). From this set, we randomly hold out 20% of the compositions for testing ($N_{\text{test}} = 200$) and train on the remaining 80% ($N_{\text{train}} \approx 800$). Importantly, while the model observes all primitives individually during training, it never sees the specific primitive combinations used at test time.

**Baseline and methods.** We fine-tune a Mistral-7B model using LoRA and compare three inference strategies: (1) a **Few-Shot Baseline**, which predicts the query output from three in-context examples; (2) **RTE**, which searches over candidate primitive pairs at test time and selects the decomposition that best explains the few-shot context; and (3) an **Oracle Bound**, in which the model is explicitly provided with demonstrations from the ground-truth atomic functions composing the target transformation.

## E. Architecture and Training Details

In this section, we detail the neural network architectures and training hyperparameters used for the synthetic experiments. All models were implemented in PyTorch.

## E.1. Figure 1: Projectile Motion Details

The motivating example in Figure 1 utilizes a simplified experimental setup distinct from the main benchmarks to strictly isolate the extrapolation behavior.

**Physics Simulation.** Trajectories are generated using the standard projectile motion equation without air resistance. The height $y$ at horizontal distance $x$ is given by:

$$y = x \tan(\theta) - \frac{gx^2}{2v_0^2 \cos^2(\theta)}$$

where $g = 9.81$.

- **Training Domain:** Velocity $v \in [30, 60]$, Angle $\theta \in [40°, 50°]$.

- **Target Task:** Velocity $v = 65$ (Out-of-Support), Angle $\theta = 45°$.

**Model Architectures.** Unlike the deeper networks used in the main results, the models for Figure 1 are shallow MLPs designed to test raw function approximation capacity.

- **Inductive Model:** A standard feed-forward network mapping $(x, v, \theta) \to y$.

    MLP: $[3, 64, 64, 1]$ with ReLU activations.

- **Transductive Model:** A relational network mapping $(x, y_{\text{anc}}, \Delta v, \Delta \theta) \to y_{\text{target}}$.

    MLP: $[4, 64, 64, 1]$ with ReLU activations.

**Training.** Both models were trained for 10,000 steps using the Adam optimizer (Kingma & Ba, 2015) with a learning rate of $5 \times 10^{-3}$. In every step, a batch of 64 trajectories was sampled.

## E.2. MLP Backbones

Across all synthetic experiments, we utilize a standardized Multi-Layer Perceptron (MLP) construction helper. We denote an MLP with layer sizes $[d_{in}, h_1, \ldots, h_k, d_{out}]$ as a sequence of linear transformations interleaved with ReLU activations.

### E.2.1. COMPOSITION AND LENGTH EXTRAPOLATION MODELS

For the *Length* and *Composition* experiments, we utilize a deeper capacity network to handle the complexity of polynomial and combinatorial arithmetic.

- **Structure:** The network is composed of two distinct feature extraction blocks followed by an output head.

    - **Block 1:** $[D_{in}, 256, 256, 128]$
    - **Block 2:** $[128, 128, 64]$
    - **Output Head:** $[64, 64, D_{out}]$

- **Activations:** ReLU is used between all hidden layers. The output layer is linear.

- **Input Dimension ($D_{in}$):**

    - **Composition Model:** $3L + L/2$ (Input $x$, component $y_f$, component $y_g$, and context head $y_{context}$).
    - **Length Model:** $2L + L/2 + 1$ (Input $x$, anchor $y_{prev}$, context head $y_{context}$, and coefficient $c_{new}$).

*Table 8.* Training Hyperparameters for Synthetic Experiments.

| EXPERIMENT | BATCH SIZE | LEARNING RATE | EPOCHS/ITERS |
|---|---|---|---|
| PARAMETER EXTRAPOLATION | 4096 | $1 \times 10^{-3}$ | 200 EPOCHS |
| LENGTH EXTRAPOLATION | 1024 | $1 \times 10^{-3}$ | 100 EPOCHS |
| COMPOSITION EXTRAPOLATION | 1024 | $1 \times 10^{-3}$ | 100 EPOCHS |
| FIGURE 1 (PROJECTILE) | 64 | $5 \times 10^{-3}$ | 10,000 STEPS |

### E.2.2. PARAMETER EXTRAPOLATION MODELS

For the *Parameter* extrapolation experiments, we utilize the architecture defined as follows.

- **Structure:**

    - **Block 1:** $[D_{in}, 64, 64, 16]$
    - **Block 2:** $[16, 64, 64, 16]$
    - **Output Head:** $[16, 128, D_{out}]$

- **Input Dimension:** $D_{in}$ varies based on the context window but generally includes the query $x$, the values of two reference anchors $y_1, y_2$, the sparse target context $y_{head}$, and the scalar distances $d_1, d_2$ in embedding space.

### E.3. Optimization Hyperparameters

All models were trained using the Adam optimizer. The specific hyperparameters derived from the source code are listed below:

For the Parameter Extrapolation experiments, we additionally apply a weight decay of $1 \times 10^{-4}$.

## F. Function Family and Dataset Specifications

### F.1. Parameter Extrapolation Families

The parameter extrapolation experiments rely on splitting continuous parameter spaces into disjoint regions: Source 1 ($F1_1$), Source 2 ($F1_2$), and Extrapolation ($F2$). The specific ranges used to generate the data are:

- **Quadratic:** $f(x) = ax^2 + bx + c$.

    - Shift Parameter $a$: $F1_1 \in [0.5, 1.5]$, $F1_2 \in [1.5, 2.5]$, $F2 \in [2.5, 3.5]$.
    - Fixed Parameters: $b, c \in [-2.0, 2.0]$.

- **Exponential:** $f(x) = \alpha(e^{\beta(x-10)} - 1) + c_0$.

    - Shift Parameter $\beta$: $F1_1 \in [0.05, 0.1]$, $F1_2 \in [0.1, 0.15]$, $F2 \in [0.15, 0.20]$.
    - Fixed Parameters: $\alpha \in [1.0, 2.0]$, $c_0 \in [-1.0, 1.0]$.

- **Cubic:** $f(x) = \frac{a_3}{400}(x - 10)^3 + \frac{b_3}{10}(x - 10) + c_3$.

    - Shift Parameter $a_3$: $F1_1 \in [0.5, 1.5]$, $F1_2 \in [1.5, 2.5]$, $F2 \in [2.5, 3.5]$.
    - Fixed Parameters: $b_3 \in [2.0, 4.0]$, $c_3 \in [-2.0, 2.0]$.

- **Trend + Wave (Sin/Tri):** $f(x) = sx + A \cdot \text{wave}(x)$. The wave component is defined with a fixed period $P = 20/7 \approx 2.86$. The functional form depends on the specific wave type:

$$\text{wave}(x) = \begin{cases} 0.5 \sin\left(\frac{2\pi x}{P}\right) & \text{if Sin-Trend} \\ 2\left|2\left(\frac{x}{P} - \lfloor\frac{x}{P}\rfloor\right) - 1\right| - 0.5 & \text{if Tri-Trend} \end{cases} \tag{15}$$

    - Shift Parameter $s$ (Slope): $F1_1 \in [0.1, 0.2]$, $F1_2 \in [0.2, 0.3]$, $F2 \in [0.3, 0.4]$.
    - Fixed Parameter $A \in [0.5, 1.0]$.

## F.2. Length Extrapolation (Polynomials)

The polynomial tasks are defined on the domain $x \in [-5, 5]$.

- **Coefficient Scaling:** To maintain numerical stability as degree $d$ increases, coefficients $c_d$ are sampled from a scaled uniform distribution:

$$c_d \sim \mathcal{U}(-2.0, 2.0) \times \frac{1}{5^d}$$

  This ensures that the contribution of the term $c_d x^d$ remains $O(1)$ within the domain.

- **Normalization:** When the coefficient $c_{new}$ is passed as input to the neural network (in the Recursive Oracle), it is multiplied by $5^d$ to normalize it back to the range $[-2, 2]$.

## F.3. Composition Combinations

The composition tasks involve three primitives: Polynomial (Poly), Sinusoid (Sin), and Hyperbolic Tangent (Tanh).

- **Training Pairs (Seen):** (Poly, Poly), (Sin, Sin), (Tanh, Tanh), (Poly, Sin), (Sin, Tanh), (Tanh, Poly).

- **Testing Pairs (Unseen):** (Sin, Poly), (Tanh, Sin), (Poly, Tanh).

For example, the model observes $\text{Poly}(\text{Sin}(x))$ during training but must predict $\text{Sin}(\text{Poly}(x))$ at test time.

# G. Latent Method Implementation Details

In the latent experiments (Section 3), we do not provide ground truth labels. Instead, we infer structure using Task2Vec embeddings and a Decomposer network.

## G.1. Task2Vec Universal Probe

To generate task embeddings $\hat{\theta}$, we utilize a lightweight "Universal Probe" rather than a large pre-trained model for the synthetic experiments.

- **Architecture:** A small MLP $[1, 64, 64, 1]$.

- **Procedure:**
    1. For a given task support set $D_\tau = \{(x_i, y_i)\}$, we initialize the probe with Xavier Uniform weights (Glorot & Bengio, 2010).
    2. We train the probe on $D_\tau$ for a short horizon (50 steps) to adapt it to the function geometry.
    3. We extract the gradients of the weights (or the weights themselves in the simplified implementation) as the raw embedding vector.
    4. We apply PCA (fitted on the training library) to project this high-dimensional vector to a compact $z$-dimension (e.g., 16).

## G.2. The Decomposer Network

For discrete regimes (Length and Composition), we train a Decomposer to map from the observed task $(x, y)$ to the latent components.

- **Permutation Invariance (Sorting Trick):** The input to the Decomposer is a set of points $\{(x_i, y_i)\}$. To ensure the network is invariant to the order of points, we explicitly sort the input tensors based on the $x$-coordinates before feeding them into the MLP.

$$\text{idx} = \text{argsort}(x); \quad x_{sorted} = x[\text{idx}]; \quad y_{sorted} = y[\text{idx}]$$

- **Architecture:** The sorted $x$ and $y$ vectors are concatenated and passed through an MLP:

$$[2L, 256, 128, 64]$$

  The output is split into heads predicting the embeddings of the components (e.g., $z_f$ and $z_g$ for composition).

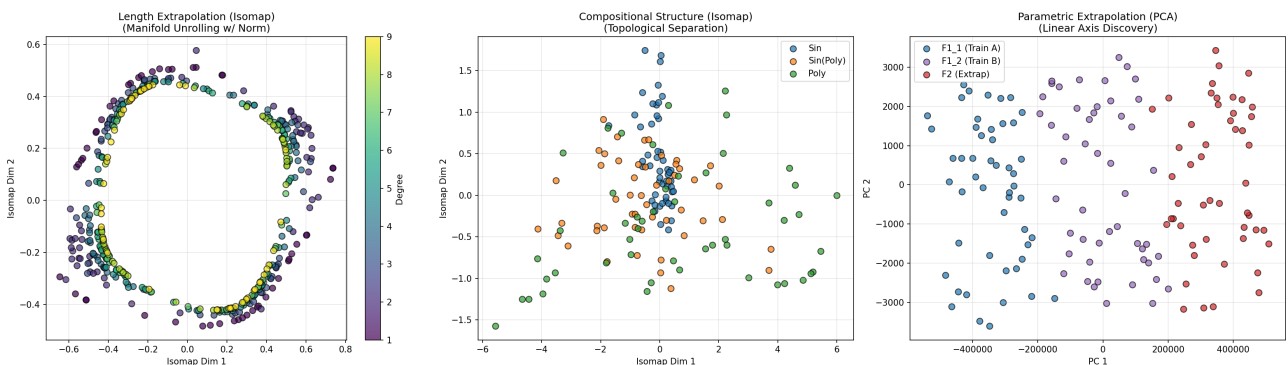

*Figure 7.* **The Geometry of Task Manifolds.** Visualizations of the gradient embeddings of three function families. **Left:** Polynomials of degrees 1–9 form concentric rings, where higher degree polynomials lie closer to the center. **Center:** Sin functions form the center of the manifold with the compositions just outside and the polynomials outside that. **Right:** Parametric shifts in coefficient space manifest as linear translations in PCA space.

## G.3. Inference and Verification

At test time, the process is as follows:

1. **Embedding:** The Decomposer predicts the latent vectors for the constituents (e.g., $z_{pred}$).

2. **Retrieval:** We use a Nearest Neighbor search (Euclidean distance) against the pre-computed embeddings of the training library to retrieve top-10 candidate functions.

3. **Verification (Re-Ranking):** For each candidate decomposition, we construct the hypothesized function (e.g., $f_{cand} \circ g_{cand}$) and evaluate its MSE on the provided sparse context (Head).

4. **Selection:** The decomposition with the lowest context MSE is selected as the anchor/transformation for predicting the query points (Tail).

# H. Visualizing the Latent Space

To validate the hypothesis that functional families form structured manifolds, we visualize the geometry of the task embeddings $\hat{\theta}$. We employ the probe-based embedding strategy described in Section G to project functions into a latent metric space.

## H.1. Methodology

We utilize a lightweight "Universal Probe" architecture, defined as an MLP $[1 \rightarrow 128 \rightarrow 128 \rightarrow 1]$ with ReLU activations. The visualization pipeline proceeds as follows:

1. **Pre-training:** For each experiment, the probe is pre-trained via standard regression on the family of curves to ensure the weights reside in a relevant region of the loss landscape.

2. **Gradient Extraction:** For each specific function instance $f_i$, we compute the gradients of the probe parameters with respect to the loss on $f_i$. These gradients serve as the high-dimensional embedding $\hat{\theta}_i \in \mathbb{R}^d$.

3. **Dimensionality Reduction:** We project the high-dimensional gradients into $\mathbb{R}^2$ using Isomap (for non-linear manifolds) or PCA (for linear shifts).

## H.2. Manifold Analysis

We conduct three experiments to verify that the embeddings capture the specific structural properties required for the extrapolation regimes defined in Section 2.

**1. Length Extrapolation (The Complexity Manifold).** We generate polynomials of degrees $d \in \{1, \ldots, 9\}$. Crucially, we normalize each function to unit energy ($L_2$ norm) to isolate the shape complexity from magnitude. We utilize Isomap ($n_{neighbors} = 15$) to unroll the intrinsic geometry. As shown in Figure 7 (Left), the tasks arrange themselves into a circular trajectory. The embedding space successfully recovers the ordinal nature of "Length": degree $d$ is adjacent to degree $d - 1$ and $d + 1$, even though the model was never provided with degree labels. This supports the *Length Extrapolation* hypothesis: increasing complexity is a traversable path in the task space.

**2. Compositional Structure (Topological Separation).** We analyze three function classes: $f(x) = \text{Poly}(x)$, $f(x) = \sin(x)$, and the composition $f(x) = \sin(\text{Poly}(x))$. Using Isomap ($n_{neighbors} = 20$), we observe distinct topological clustering (Figure 7, Center). The simple atomic functions (Poly and Sin) form their own loose clusters, while the composite functions ($\sin \circ \text{Poly}$) form a dense, separate region. This confirms that the embedding is sensitive to the frequency modulation introduced by composition, a necessary condition for the *Compositional Extrapolation* regime.

**3. Parametric Extrapolation (Linear Axis Discovery).** We generate quadratic functions $f(x) = ax^2 + bx + c$ where the curvature parameter $a$ is shifted across three disjoint regions: Train A ($a \in [0.5, 1.5]$), Train B ($a \in [1.5, 2.5]$), and Extrapolation ($a \in [2.5, 3.5]$). Because the shift is a linear transformation of the coefficients, we utilize Principal Component Analysis (PCA). Figure 7 (Right) demonstrates that the tasks lie on a linear manifold. The shift from Train A to Train B defines a vector $\vec{v}$. Extrapolating this vector $\vec{v}$ from Train B lands exactly in the Extrapolation region (Red), validating the *Parameter Extrapolation* vector algebra.

# I. LLM Task Details

## I.1. Sparse Parity Setup

**Model Optimization.** We utilize `Qwen/Qwen2.5-7B-Instruct` as the base model. We apply LoRA fine-tuning with rank $r = 16$, $\alpha = 32$, and dropout $0.05$ targeting the query, key, value, and output projections. The model is trained for 600 steps with a batch size of 1 and gradient accumulation of 16 (effective batch size 16). We use the AdamW optimizer (Loshchilov & Hutter, 2019) with a learning rate of $2 \times 10^{-4}$ and a cosine decay schedule with a 50-step warmup.

**Prompt Formats.** We utilize specific prompt templates for the Baseline and the Extender (Transductive) modes.

- **Baseline Prompt:**

```
Instruction:  Compute the Output.\n###\n
In:  0 1 0 1 1 0 0 1 | Out:  1
...
In:  1 1 0 0 1 0 1 0 | Out:
```

- **Extender Prompt (Relational):** The model is provided with the input string, the output of the *Anchor* task (Src), and the value of the specific bit at the transformation index (BitVal).

```
Instruction:  XOR the Source with the BitVal.\n###\n
In:  0 1 0...  | Src:  0 | BitVal:  1 -> Tgt:  1
In:  1 1 0...  | Src:  1 | BitVal:  1 -> Tgt:  0
```

**Search Procedure.** At test time, we are given a few-shot context $D_{test}$ for a task with hidden mask $S$ ($|S| = 6$). We perform a brute-force search over the training library to find the decomposition.

1. We iterate over all masks $S_{cand}$ in the training set (where $|S_{cand}| \in [2, 5]$).

2. We iterate over all possible bit indices $k \in [0, 7]$.

3. We construct the *Extender Prompt* using $S_{cand}$ as the source and $k$ as the bit value.

4. We compute the likelihood of the ground truth labels in $D_{test}$ under this configuration.

5. We select the pair $(S_{cand}, k)$ with the lowest loss to predict the query.

## I.2. CodeIO Primitives and Training

The CodeIO experiments use a library of atomic string transformation functions implemented in Python. The model interacts with these functions via few-shot input-output pairs. The complete set of primitives derived from `ATOMIC_FUNCS` is listed below:

- **Permutations:** `reverse`, `shuffle`, `sort_chars`, `rotate_1`, `rotate_3`, `mirror` (append reverse), `reverse_words`, `interlace_self_rev` (interlace string with its reverse).

- **Filters:** `remove_vowels`, `filter_alpha` (keep only alphabetic), `compress_repeats` (aaabb → ab), `verify_even` (truncate if odd length).

- **Mappings:** `shift_1` (Caesar +1), `shift_13` (ROT13), `vowel_to_num`, `alternate_case`, `upper_case`, `lower_case`.

- **Expansions/Formatting:** `duplicate_chars`, `repeat_2` (string × 2), `loop_concat_2`, `add_prefix_x`, `add_suffix_y`, `fancy_brackets` (wrap chars in ≪≫), `sep_dash` (insert dashes).

- **Complex/Recursive:** `recursive_interlace`, `force_digit` (back-chaining search to add a digit), `force_palindrome`, `concat_self`, `while_rotate`.

**Implementation Details.** We utilized the Unsloth library for efficient fine-tuning.

- **Model:** `mistral-7b-instruct-v0.2-bnb-4bit`.

- **LoRA Config:** Rank $r = 32$, Alpha $\alpha = 16$, Targets: [`q_proj`, `k_proj`, `v_proj`, `o_proj`, `gate_proj`, `up_proj`, `down_proj`].

- **Training:** The model was trained for 4 epochs with a learning rate of $2 \times 10^{-4}$ and a batch size of 2 (gradient accumulation steps = 4) with the AdamW optimizer.

- **Search:** During the search phase, we evaluated candidates using a batch size of 8. The scoring metric was the sum of Cross-Entropy Loss over the tokenized completions of the 3 support examples provided in the prompt.

## I.3. Prompting and Input Representation

To strictly evaluate the model's ability to reason compositionally rather than retrieve memorized definitions, we employ a strict masking and formatting protocol.

**Function Masking.** All atomic primitives are anonymized to decouple the reasoning process from linguistic semantic priors. We construct a global registry that maps descriptive function names to arbitrary identifiers (e.g., `reverse` → `Func_15`). This ensures that the model must infer the function's mechanics solely from the provided input-output examples, rather than relying on the token "reverse" to trigger a pre-learned behavior.

- **Implementation:** The mapping is deterministic based on the sorted alphabetical order of the function names (e.g., 'ATOMIC_FUNCS' keys).

- **Goal:** The model learns to treat `Func_XX` as a variable representing a specific transformation rule, rather than a semantic label.

**Context Alignment.** When providing context for a candidate primitive (e.g., `Func_15`), we generate dynamic few-shot examples. Crucially, we force the context examples to align with the current task inputs. If the current task involves transforming the string "axiom", the context block for the atomic functions will explicitly show how they act on "axiom".

$$\text{Input}_i \in \text{Task} \implies \text{generate}(\text{RefFunc}, \text{Input}_i) \in \text{Prompt}$$

This "Context Alignment" ensures the model has the necessary local information to compute the composition step-by-step, mimicking a working memory retrieval where the relevant intermediate states are brought into focus.

**Prompt Template.** The final prompt presented to the model follows a strict hierarchy: (1) Reference Block (definitions of the atomic candidates), (2) Support Set (few-shot examples of the composite task), and (3) The Query. An illustrative example of the prompt structure is provided below:

```
You are provided with the following Reference Functions.  Study their behavior:
-- Func_15 --
In:  apple Out:  elppa
In:  orange Out:  egnaro

-- Func_22 --
In:  elppa Out:  ELPPA
In:  egnaro Out:  EGNARO

Now, use the Reference Functions above to solve the Target Task:
In:  apple Out:  ELPPA
In:  orange Out:  EGNARO

### Query:
In:  banana Out:
```

In the **Baseline** condition, the "Reference Functions" block is removed, forcing the model to rely solely on the inductive pattern in the "Target Task" section. In the **Compositional Search** condition, the Reference Block is populated with the top candidates retrieved by the search algorithm.

## J. Methodology for Constructing LLM Geometry

For the LLM experiments (Section 4), we avoided computing the geometry of the tasks because of the long inference time. However, we do still show that such a geometry can be computed.

**1. Parameter-Efficient Proxies (LoRA).** Instead of computing the Fisher Information Matrix (FIM) with respect to the full model weights $\omega$, we inject Low-Rank Adapters (LoRA) into the attention and MLP blocks of the frozen LLM. We define the task parameter vector $\theta$ solely as the flattened concatenation of the adapter matrices $A$ and $B$.

$$\theta_{task} = \text{concat}(\text{vec}(A_l), \text{vec}(B_l)) \quad \forall l \in L_{active}$$

This reduces the dimensionality of the embedding space from billions to thousands, focusing the metric solely on the parameters responsible for *adapting* to the new task.

**2. Layer Selection** A key insight for probing LLM tasks is that not all layers of a Transformer contribute equally to algorithmic processing. Early layers often attend to low-level syntax and positional encoding, while final layers focus on vocabulary projection and formatting. The semantic "reasoning", the actual transformation of information, typically occurs in the middle depth.

If one computes the Fisher Information Matrix (FIM) over the entire network, the high-magnitude gradients from the syntactic layers (which process the input format) often drown out the subtle gradients of the reasoning layers.

**Implementation:** We designate layers 10 through 32 (of a 36-layer GPT2-Large (Radford et al., 2019) backbone) as the **Logic Core**. We restrict the probe to only compute embeddings derived from parameters in this range.

**3. Layer-Wise Isotropic Normalization** Gradient magnitudes in deep networks are not uniform; they vary significantly by depth due to the dynamics of backpropagation. Without normalization, a single layer with high gradient variance can dominate the Euclidean distance in the embedding space, rendering the metric insensitive to changes in other layers.

**Implementation:** We apply **Layer-Wise L2 Normalization**. We treat the parameters of each layer $l$ as a distinct vector $F_l$. We normalize each layer independently before concatenation:

$$\theta_{emb} = \bigoplus_{l=10}^{32} \frac{F_l}{\|F_l\|_2 + \epsilon}$$

This forces every layer in the Logic Core to contribute equally to the final geometric representation, ensuring that the embedding captures the multi-step reasoning process rather than just the single most active layer.

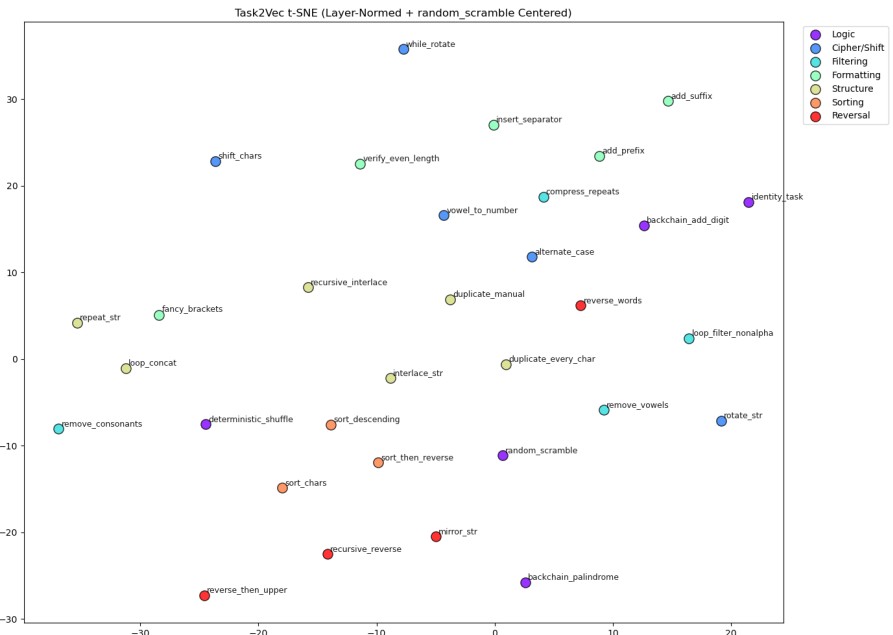

*Figure 8.* **Geometry of CodeIO (2-Dim TSNE.)** Even with a fairly small model, clear clusters form corresponding to semantic similarities between tasks.

To apply these general principles to the specific domain of String Manipulation (`CodeIO`), we employed two domain-specific techniques.

**Tokenization: Space-Delimitation.** Byte-Pair Encoding (BPE) introduces artifacts for character-level tasks (e.g., "reverse" becomes dependent on how a word is fragmented). To force the probe to learn the general algorithm rather than specific token memorization, we space-delimit all inputs (e.g., `"apple"` → `"a p p l e"`). This ensures the gradient signal reflects the character manipulation logic.

**Baseline Subtraction: Scramble Centering.** Even with the Logic Core, embeddings often contain a shared "background" vector representing the general effort of processing the input distribution. To remove this, we define a "Null Hypothesis" task: `random_scramble`, which permutes inputs randomly without logic. We compute the final embedding by subtracting this baseline:

$$\theta_{final} = \theta_{raw} - \theta_{scramble}$$

This operation emoves the generic "string processing" signal, leaving only the residual vector that encodes the specific algorithmic structure (e.g., *Sorting* vs. *Reversing*).

**Results** Employing the previous techniques, we were able to produce Figure 8 after applying TSNE (van der Maaten & Hinton, 2008) for dimensionality reduction. The clustering of the sorting tasks, formatting tasks, and reversal tasks demonstrate meaningful structure was observed. The geometry here is still imperfect. We hypothesize this is because the quality of the embeddings correspond to the ability of the model to be able to solve the task; otherwise the LoRA weights don't correspond to anything meaningful. We used a small model in GPT2-Large as opposed to the much larger Mistral model we were using earlier to save on compute time, which degrades the quality of the embeddings.

## K. Ablation Studies

To validate that the performance reported in Section 3 stems from the geometric exploitation of the task manifold rather than stochastic noise, we perform a series of ablation studies. We isolate the contributions of **Anchor Selection** and **Transformation Structure**.

*Table 9.* **Parameter Extrapolation Ablations (Quadratic).** Comparison of MSE on the extrapolation region F2. The results demonstrate that proximity on the manifold (Near vs. Far) is the dominant factor in success.

| METHOD | MSE | 95% CI |
|---|---|---|
| INDUCTIVE BASELINE | 1.05E5 | 4.6E3 |
| **STANDARD TRANSDUCTIVE (NEAR)** | **1096.9** | 169.1 |
| *Ablations on Topology* | | |
| FAR ANCHORS ($F1_1$) | 2325.5 | 357.9 |
| RANDOM ANCHORS | 1322.8 | 225.9 |
| *Ablations on Shift* | | |
| ZERO SHIFT | 1158.6 | 108.4 |

### K.1. Parameter Extrapolation

To validate the hypothesis that extrapolation relies on traversing a structured manifold, we analyze the Quadratic family defined in Section F. The task is to predict curvature parameters in the Extrapolation region ($a \in [2.5, 3.5]$). We compare the standard Transductive model against the Inductive baseline and three structural ablations:

- **Far Anchors:** We force the model to retrieve anchors from the distant training region $F1_1$ ($a \in [0.5, 1.5]$) instead of the adjacent region $F1_2$ ($a \in [1.5, 2.5]$).

- **Random Anchors:** We retrieve random anchors from the training set, ignoring the embedding geometry.

- **Zero Shift:** We provide the nearest neighbor anchor but mask the shift parameters ($d_1, d_2 = 0$), forcing the model to rely solely on the anchor's shape.

**Results.** As shown in Table 9, the Inductive Baseline fails catastrophically ($MSE \approx 1.05\text{e}5$), confirming that standard networks cannot generalize curvature outside their training support.

The ablations highlight the importance of **Topological Fidelity**. The *Standard* model (using nearest neighbors in $F1_2$) achieves an MSE of 1096. When forced to use *Far Anchors*, error doubles to 2325. This confirms that task extrapolation is local; performance degrades as the distance on the manifold increases. *Random Anchors* (1322) perform better than Far Anchors because the random selection includes tasks from the proximal $F1_2$ region, whereas the Far ablation explicitly excludes them.

Finally, the *Zero Shift* ablation (1158) performs comparably to the Standard model. This suggests that for simple families like Quadratics, identifying the correct "Base Shape" (Anchor) accounts for the majority of the gain, while the learned shift vector provides fine-grained refinement.

### K.2. Length Extrapolation

To understand the mechanics of the transductive framework in the length extrapolation regime, we perform ablation studies on the Degree 9 polynomial extrapolation task. We isolate how performance degrades as anchor selection deteriorates. The results are summarized in Table 10.

**Sensitivity to Anchor Retrieval.** The performance of the system is heavily dependent on the quality of the retrieved anchor. When the model is forced to use a *Random Anchor* from the library, the error increases dramatically ($MSE \approx 4.67$). This confirms that the Extender network requires a structurally relevant base function to apply the transformation. However, the system is robust to minor retrieval errors: using the *Rank-3* candidate instead of the best-verified anchor only results in a marginal performance drop ($0.73 \to 0.80$).

**Importance of the Learned Shift.** We evaluate a *Zero Shift* condition where the model retrieves the nearest neighbor anchor and the transformation parameter $c_{new}$ is set to zero. This isolates the utility of sophisticated search in this context. While this still outperforms the Inductive Baseline (0.806 vs 1.382), it underperforms the anchor search process. This indicates that while the anchor provides the necessary "backbone," the learned relational shift is useful for modelling.

**The Retrieval Gap.** The gap between the Standard Transductive model (0.732) and the Oracle (0.351) suggests that even

*Table 10.* **Length Extrapolation Ablations (MSE on Degree 9).** We evaluate the sensitivity of the system to anchor retrieval and the necessity of the learned shift coefficient.

| METHOD | MSE $\pm$ 95% CI |
|---|---|
| INDUCTIVE BASELINE | $1.3823 \pm 0.1596$ |
| **TRANSDUCTIVE (STANDARD)** | $\mathbf{0.7322 \pm 0.0793}$ |
| *Ablations on Retrieval* | |
| RANK-3 ANCHOR | $0.8032 \pm 0.0936$ |
| RANDOM ANCHOR | $4.6664 \pm 0.4020$ |
| *Ablations on Transformation* | |
| ZERO SHIFT (NEURAL) | $0.8060 \pm 0.1022$ |
| *Upper Bound: Oracle* | $0.3510 \pm 0.0399$ |

with verification, the retrieval of the "correct" Degree 8 predecessor is the primary bottleneck. Improving the fidelity of the Task2Vec manifold or utilizing more sophisticated search strategies remains a promising direction for narrowing this gap.

### K.3. Compositional Extrapolation

We evaluate the Compositional regime by degrading the retrieval and scoring mechanisms. We analyze whether the performance gain stems from the semantic structure of the embedding space or the transductive verification step (checking the reconstruction loss on the context set).

We compare the **Standard Transductive** model (which retrieves top-$k$ candidates and selects the best fit) against three variations:

- **Oracle:** The model is provided with the ground-truth constituents $f$ and $g$. This represents the theoretical upper bound of the Composition Operator $\Psi_{comp}$.

- **Random Constituents:** The Operator is fed random functions from the library instead of the retrieved constituents. This tests whether the operator actually uses the provided functions to simulate the target?

- **No Verification:** We rely solely on the Decomposer network's prediction. The nearest neighbor in embedding space is selected immediately as the anchor, skipping the physics-based verification step (calculating MSE on the sparse context).

**Results.** As shown in Table 11, the **Standard Transductive** model (0.58) outperforms the Inductive Baseline (0.65). The **Oracle** results (0.50) confirm that if the correct components are identified, the operator can approximate the composite function with high precision.

Crucially, the **Random Constituents** ablation results in a catastrophic error increase (1.11). This confirms that the operator $\Psi_{comp}$ functions as a faithful simulator: if provided with incorrect primitives (wrong keys), it synthesizes a divergent target function.

Furthermore, the **No Verification** ablation (0.68) performs slightly worse than the inductive baseline. This highlights a critical insight: the latent embedding space provides a *coarse* search region, but the Decomposer is not precise enough on its own. The improvement in the Standard Transductive method is driven by the **Test-Time Verification** step, using the sparse context points to filter the neural retrieval results.

*Table 11*. **Composition Extrapolation Ablations.** Comparison of MSE ($\pm$ 95% CI). The failure of "No Verification" demonstrates that search-based re-ranking is necessary to outperform the baseline. "Random Constituents" confirms the model actively relies on the retrieved components.

| METHOD | MSE | CI |
|---|---|---|
| INDUCTIVE BASELINE | 0.6545 | 0.0949 |
| **STANDARD TRANSDUCTIVE** | **0.5792** | **0.0815** |
| *Ablations* | | |
| NO VERIFICATION (TOP-1) | 0.6773 | 0.0990 |
| RANDOM CONSTITUENTS | 1.1109 | 0.1122 |
| *Upper Bound* | | |
| ORACLE (GROUND TRUTH) | 0.4968 | 0.0690 |

