# OpenReview forum: "Learning to Extrapolate to New Tasks: A Relational Approach to Task Extrapolation"
_ICML.cc/2026/Conference — ICML 2026 regular_

### Official Review · Reviewer_Q2gz · 2026-03-09

**Soundness:** 3
**Presentation:** 4
**Significance:** 3
**Originality:** 3
**Overall Recommendation:** 4
**Confidence:** 3

**Summary:**

This paper presents an task extrapolation methods that learns to extend model's task handling capacity through combination of a existing task, a task transduction descriptor, and a transduction function, where the transduction function is trained from tasks that the model can support currently. During inference time, the algorithm needs to search for a proper existing task and a task transduction descriptor using trained transduction function such that achieve new task handling. Experiments shows the method works for both classic machine learning models and the LLM models. In particular, for LLM, the task transduction descriptor is part of prompt. And the transduction function is the fune-tuned LLM model.

**Compliance With Llm Reviewing Policy:**

Affirmed.

**Final Justification:**

I think the authors answered my question and I will keep my score based on the discussion.

**Key Questions For Authors:**

Does this method require maintain two LLM models at the same time? one serves for known tasks handling and another serves for task transduction?

**Limitations:**

I don't see societal impact limitation of this work.

**Strengths And Weaknesses:**

Strengths
1. The paper aims to tackle a very important problem in current LLMs, where a lack of task extrapolation limits their utility in many applications with sparse training data. The proposed approach is a promising direction to explore. In particular, the extrapolation mechanism combining an anchor task and a trainable transduction function is novel to me.
2. The paper is well-written and provides good intuition as to why the proposed method should work. While the description of adapting the LLM is not entirely comprehensive, the paper clearly delivers the general idea.

Weaknesses
1. I remain skeptical about the effect of the proposed method on LLMs, since fine-tuning an LLM to instill new behaviors is generally not feasible without sufficient data. Furthermore, the experiments are too simple and not very convincing, given that LLMs may already have the ability to make such predictions with proper prompting.
2. The method requires searching for an anchor task and a transduction descriptor at inference time, which is highly impractical. Even if this were possible given limited access to the training data, the transduction descriptor does not seem very generalizable and may require further adaptation for truly unseen or unknown new tasks.

---

> ### Author Rebuttal · Authors · 2026-03-28
>
> # Thank you for appreciation of our work and keen questions
>
> Thank you for your constructive feedback. We are glad you enjoyed our work and acknowledge the importance of the problem that RTE solves. We appreciate the opportunity to respond to the weaknesses you have identified.
>
> Before addressing the specific questions, we note that **the primary focus of this paper is not on LLMs specifically but on a broader problem in ML**. Neural networks tend to saturate outside of the domain due to ReLU activations - RTE is an attempt to address this phenomenon, which arises broadly across neural network applications.
>
> ## 1. Does RTE maintain 2 models?
>
> Yes, one model generates the Task2Vec geometry and one model is used for transduction. **These models need not be the same size or architecture.** For instance, in Appendix J, we computed embeddings with GPT2-Large and performed transduction with a much larger Mistral-7B model. However, if one chooses a different search strategy at test time that does not rely on embeddings, maintaining two models is not necessary. In practice, we found that maintaining two models did not introduce significant engineering overhead.
>
> ## 2. Does RTE instill new behaviors?
>
> You rightly point out that teaching an LLM entirely new behaviors from sparse data is infeasible. RTE does not attempt this. Instead, it mines latent structure and provides it to the LLM to make its job easier. **RTE teaches the model how to algebraically compose already learned behaviors.** By reducing an out-of-support (OOS) problem into an out-of-combination (OOC) problem, the model relies on structural mechanisms it has already robustly learned during training, but RTE guides it to apply these mechanisms correctly. To address the concern that LLMs might already achieve this with proper prompting, our experiments explicitly test this boundary. As shown in Sections 4.1 and 4.2, standard few-shot prompting baselines fail catastrophically on these extrapolation tasks (e.g., dropping to 29.7\% on the CodeIO benchmark). Standard prompting is insufficient for these specific structural leaps; the explicit relational decomposition provided by RTE is required to recover performance and the model needs to be taught how to use these relationships.
>
>  ## 3. Feasibility of Search and Generalization of RTE
>
>  We agree that exhaustive search at inference time is computationally expensive. To make this practical, we utilize Amortized Discrete Search (detailed in Section 2.4.2, Strategy B). **By training a lightweight decomposer network to predict a distribution over plausible anchor-transformation pairs, we restrict the search space to only the top-$k$ candidates.** This drastically reduces the inference-time cost while maintaining accuracy, making the approach computationally tractable. Also, test-time compute is increasingly the norm in ML - see Snell et. al, 2024.
>
>  Furthermore, you raise a point about the generalizability of the transduction descriptor. One natural generalization is that RTE natively supports higher-order compositions (e.g., $f(g(h(x)))$) through recursive application of the relational operator: $\Psi(\Psi(x, f_{anc}, g_{trans}), h_{trans})$. The limitation in this case is not the operator's capacity, but the combinatorial explosion during the decomposition phase. We will add a discussion paragraph detailing this mechanism and noting how heuristics like beam search or learned decomposer networks can maintain tractable inference for deeper compositions.
>
>  As we understand it, another dimension of your critique is that transformations can be vague or difficult to explicitly enumerate in words,  making them unintuitive to parameterize. RTE’s core premise is that we can make these problems tractable by grounding these latent transformations into quantitative proxies (such as the Task2Vec embeddings demonstrated in our latent experiments). If no quantitative parameterization of a transformation exists, then RTE would not apply. We will clarify this scope in the introduction.
>
>  *References*
>  *- Snell, Charlie, et al. "Scaling LLM Test-Time Compute Optimally can be More Effective than Scaling Model Parameters.", The Thirteenth International Conference on Learning Representations."*

---

> > ### Author Rebuttal · Reviewer_Q2gz · 2026-04-02
> >
> > Thank you for your reply. I will keep my score.

---

> > > ### Author Response · Authors · 2026-04-07
> > >
> > > # Thank you for your time and your thoughtful review.
> > >
> > > We are pleased that our responses have answered your questions. Thank you again for your thoughtful feedback and for confirming your evaluation.

---

### Official Review · Reviewer_JJTp · 2026-03-12

**Soundness:** 2
**Presentation:** 2
**Significance:** 2
**Originality:** 2
**Overall Recommendation:** 3
**Confidence:** 4

**Summary:**

The paper talks about interpolated vs extrapolated tasks, an approach to learn transformations wrt extrapolated tasks, and contributes a method to do this in a way that yields better empirical performance.

**Compliance With Llm Reviewing Policy:**

Affirmed.

**Final Justification:**

My concerns still remain, and thus I retain my assessment.

**Key Questions For Authors:**

Which experiments would you say most strongly highlight this extrapolation performance? Like a task where it is very distant/far from training tasks, but the model still does well at it?

**Strengths And Weaknesses:**

I think the motivation is good to view tasks as either interpolated vs extrapolated. This is sensible and a topic worthwhile to study.

I think the way that an extrapolated task is defined, formulated, implemented, and modeled might be an issue. If a task can be composed from an existing set of tasks, that is an interpolation of those tasks, if I understood correctly? Even in traditional domain adaptation literature, when a source task/distribution is transformed into a target task/distribution, it is not said that they are two different tasks, just a distribution with a style/domain applied to it. I think it is ok for the contribution of the paper that it proposes a method that makes use of intutions/insight around transformation, but the underlying theory and core intuition around extrapolation is not necessarily justified in my view. More critically, it is not clear if this method would scale to actual extrapolated tasks, where the task is compositionally non-identical to any given task in a task distribution in any way.

---

> ### Author Rebuttal · Authors · 2026-03-28
>
> # Thank you very much for your Review
>
> Thank you for your thoughtful review and for highlighting the conceptual boundary between extrapolation and interpolation. Your core intuition, that solving an extrapolated task by decomposing it into in-distribution components is a form of interpolation, aligns perfectly with the central thesis of our paper.
>
> ## Regarding the Definition of Extrapolation and Interpolation:
> We completely agree that if a model can decompose an unseen task into known transformations, the problem behaves more like interpolation. However, from the perspective of a standard inductive model (i.e., a neural network or LLM), **a target task whose descriptor lies completely entirely the training domain is a true, ill-posed out-of-support (OOS) extrapolation problem**. For instance, if the training domain is $x \in (0,10)$, standard models fail catastrophically when tested on $x=15$. But RTE would say "I've seen in my training set a shift of $\Delta x = 5$ and I've seen $x=10$, so I can apply that shift to that point and get to $x=15$". In this sense, RTE learns to convert what would typically be an extrapolation problem into an interpolation problem to make it solvable.
>
> Our core contribution is demonstrating exactly what you noted: **we can render intractable OOS extrapolation tractable by reframing it as an out-of-combination (OOC) transductive problem where both components are in-domain**. Building on the foundational insights of Netanyahu et al. (2023), we show that for an LLM, evaluating on an unseen task composition is literally going outside of its domain. The key insight is that, if you reframe the domain to this OOC setup, you can reframe an extrapolation problem into an interpolation problem. We also clarify that unlike classical domain adaptation, which often relies on simple parameter offsets, our framework captures substantially richer task-to-task relationships (such as recursive extension and functional composition). We will update the introduction to make this theoretical bridge, namely reducing extrapolation to OOC interpolation, more explicit for the reader.
>
> ## Experiments Highlighting Extrapolation
>
> To answer your question regarding which experiments highlight performance on truly distant tasks, we point to two distinct regimes where the target task is compositionally or parametrically disjoint from the training set:
>
> - *Compositional Extrapolation (CodeIO):* In the CodeIO benchmark, the model must predict the output of a composite string transformation (e.g., $T = f_b \circ f_a$). The target tasks are specifically chosen to be compositionally distinct from any task in the training distribution; while the model sees the atomic primitives during training, it never sees the specific combinations used at test time. Because this unseen combination is entirely out of domain, the standard inductive LLM baseline fails, achieving only 29.7\% accuracy and frequently hallucinating. **By reframing this extrapolation as an OOC interpolation of known primitives, RTE nearly doubles the performance to 57.3\%.**
>
> - *Parameter Extrapolation:* In our continuous regime, target parameters lie strictly outside the training support. For example, models are trained on quadratic shifts where $a \in [0.5, 2.5]$, but tested entirely outside this range on $a \in [2.5, 3.5]$. Inductive models saturate at the boundary of their training experience and fail catastrophically, yielding an MSE of roughly 115,000. **RTE successfully recovers the out-of-range curvature by anchoring to a known boundary task and applying the learned transformation, achieving an MSE of 134, roughly 3 orders of magnitude better than a standard neural network.**
>
> We believe that leveraging these relational structures to achieve high performance on strictly out-of-support tasks constitutes genuine task extrapolation. We hope this clarifies our theoretical stance and empirical results, and we thank you for the opportunity to strengthen the framing of the paper.

---

> > ### Author Rebuttal · Reviewer_JJTp · 2026-03-31
> >
> > Unfortunately I am still a little unconvinced by the premise and setup, based on my review.

---

> > > ### Author Response · Authors · 2026-04-07
> > >
> > > # Thank you for your response
> > > Thank you for raising your concern about our work. To clarify, we would like to be precise about how we use the term *extrapolation*.
> > >
> > > In this paper, we define **extrapolation as out-of-support (OOS) prediction**. In the standard prediction setting, OOS means we observe data from a function $f$ over inputs $x \in [a,b]$, and the goal is to predict $f(x')$ for inputs $x' \notin [a,b]$. In the task extrapolation setting, OOS arises at the level of functions. Specifically, we consider a mapping $F(\theta)$ that generates functions $f_\theta = F(\theta)$. We observe data from functions corresponding to parameters $\theta \in [a,b]$ (without direct access to the underlying $\theta$), and aim to make predictions about functions $f_{\theta'}$ for $\theta' \notin [a,b]$. This formulation is consistent with prior work in the literature (see also our rebuttal to reviewer RuVx).
> > >
> > > Our approach is based on three core claims about OOS problems.
> > >
> > > ## 1. OOS Problems Are Fundamentally Challenging for Neural Networks
> > > **Standard prediction models often fail when evaluated outside their training support.** This limitation is well documented in classic domain adaptation and even more substantiated in the literature for neural nets.
> > >
> > > - Ben-David, S., Blitzer, J., Crammer, K., Kulesza, A., Pereira, F., & Vaughan, J. W. (2009). A theory of learning from different domains. Machine Learning.
> > >
> > > This work shows that generalization error roughly grows with the structural divergence between training and test domains. In the OOS setting, however, divergence is not controlled and often infinite when the test domain is out of the support of the training domain. Therefore, the error is often unbounded without additional assumptions.
> > >
> > > - Xu, K., Zhang, M., Li, J., et al. (2020). How Neural Networks Extrapolate: From Feedforward to Graph Neural Networks.
> > >
> > > This paper demonstrates that neural networks tend to saturate near the boundaries of the training distribution and wildly underperform on OOS inputs.
> > >
> > > - Dziri, N., Lu, X., Sclar, M., et al. (2023). Faith and Fate: Limits of Transformers on Compositionality.
> > >
> > > This work highlights the difficulty transformers face in compositional extrapolation tasks, further illustrating the broader challenge of OOS generalization.
> > >
> > > Empirically, our results align with these findings: standard neural network baselines exhibit substantial performance degradation across all experiments (see Table 1).
> > >
> > > ## 2. Out-of-Combination (OOC) Framing Makes OOS Problems More Tractable
> > > If an OOS problem can be reframed as an Out of Combination (OOC) problem, where the target task is decomposed into an anchor and a transformation within the training support, the problem behaves more like interpolation. **This transductive perspective is well established in the literature and has been successfully applied across multiple domains:**
> > >
> > > - Netanyahu, A., Gupta, A., Simchowitz, M., Zhang, K., & Agrawal, P. (2023). Learning to Extrapolate: A Transductive Approach. The Eleventh International Conference on Learning Representations.
> > >
> > > This paper introduced the OOC view with bilinear transduction.
> > >
> > > - Segal, N., Netanyahu, A., Greenman, K. P., Agrawal, P., & Gómez-Bombarelli, R. (2025). Known Unknowns: Out-of-Distribution Property Prediction in Materials and Molecules. npj Computational Materials.
> > >
> > > This paper demonstrates the effectiveness of OOC framing for OOS prediction in materials science.
> > >
> > > - Song, Y., Lee, D., & Kim, G. (2024). Compositional Conservatism: A Transductive Approach in Offline Reinforcement Learning. The Twelfth International Conference on Learning Representations.
> > >
> > > This paper applies OOC framing successfully in offline reinforcement learning.
> > >
> > > ## 3. Empirical Results: RTE Enables OOC Framing for Task Extrapolation
> > > In the functional setting, OOC reframing is more challenging. In many cases, the underlying task parameters $\theta$ are latent, making OOC framing a lot more challenging.  Also, novel compositions are naturally OOC.
> > >
> > > Despite this, our results show that **RTE leverages these relationships to meaningfully outperform standard neural nets on OOS problems**, enabling OOC-style reasoning in task extrapolation. A particularly clear example is the quadratic setting in Table 1. We consider functions of the form: $ax^2 + bx + c$. Training data includes functions with $a \in [0.5, 2.5]$, while evaluation is on functions with $a \in [2.5, 3.5]$: a strictly OOS regime. By leveraging Task2Vec, RTE successfully reframes this problem as OOC, resulting in a **three orders of magnitude reduction in error** compared to standard approaches.
> > >
> > > ## In summary:
> > > 1. **OOS problems are inherently difficult.**
> > > 2. **OOC reduction is a principled and established strategy for addressing OOS challenges.**
> > > 3. **RTE provides a mechanism for applying OOC reduction to task extrapolation problems.**
> > >
> > > We hope this clarifies the motivation and contributions of our work. Thank you again for your thoughtful review.

---

### Official Review · Reviewer_uWJE · 2026-03-12

**Soundness:** 3
**Presentation:** 4
**Significance:** 3
**Originality:** 3
**Overall Recommendation:** 4
**Confidence:** 4

**Summary:**

The paper proposes a novel relational approach for extrapolating to unseen tasks. The central idea is to reformulate inference on unseen tasks by treating the novel task as a function (defined as a relational operator) of the input, a related anchor task, and a transformation of underlying task parameters ($\phi$). Both the relational operator and the task-parameter transformation are learned within an optimization framework using ground-truth relational mappings for training tasks. At inference time, the closest anchor task is selected, and the relevant parameter and relational transformation are applied via various strategies. The framework is demonstrated across three forms of task extrapolation: parameter extrapolation, length extrapolation, and compositional extrapolation. An extension to LLMs via relational fine-tuning is also presented. Experimental results on synthetic data and existing benchmarks demonstrate superior performance compared to standard inductive learning methods.

**Compliance With Llm Reviewing Policy:**

Affirmed.

**Final Justification:**

As noted in the rebuttal acknowledgment below, I have two key concerns and would encourage the authors to address them in the revision.

Overall, I remain positive about the work because I think this unique relational perspective on task generalization would inspire future in this direction to make these ideas more practical. For this reason, I'd like to keep my current score.

**Key Questions For Authors:**

1. In line with the weaknesses above, could you explain if the proposed approach would scale to general cases of parameter, length, and compositional extrapolation, or is limited to certain sub-classes of problems only? If yes, what conditions should those classes satisfy? More specifically, could you explain if the approach would extend to a variety of length generalization benchmarks proposed in Zhou et al. (2024a) and the compositional generalization benchmark proposed in Ramesh et al. (2024)?

2. The formulation of $\phi$ as a difference in task parameters (or as learned task embeddings) seems restrictive. Two related questions arise: (1) Are there any smoothness assumptions in the task parameter space required to theoretically justify successful extrapolation? (2) How does the framework handle settings where task parameters are not linearly transformable?

3. The concept of the relational operator is somewhat ambiguous — it is unclear whether $\Psi$ is a fixed parameter or a parameterized model consisting of multiple parameters. The notation $\Psi(x, f_i, \phi_{ij})$ suggests it takes an input, a function, and a relational transformation parameter simultaneously, yet in the LLM setting, it appears to be instantiated as a full parameterized model. Could this distinction be made more explicit and $\Psi$ be defined more formally?

4. Could the authors clarify the respective roles of $\Psi$ and $\phi$? My understanding is that $\phi$ captures shifts in parameter space while $\Psi$ captures more general functional transformations between tasks. If so, why is $\phi$ needed explicitly — couldn't $\Psi$ subsume the task *and* parameter transformation directly? This distinction seems especially unclear in the compositional generalization setting, where the role of $\phi$ is not well defined, as seen in the examples.

References
- Zhou, Hattie, et al. "What Algorithms can Transformers Learn? A Study in Length Generalization." The Twelfth International Conference on Learning Representations.
- Ramesh, Rahul, et al. "Compositional Capabilities of Autoregressive Transformers: A Study on Synthetic, Interpretable Tasks." Forty-first International Conference on Machine Learning.

**Limitations:**

Though there is no explicit discussion of limitations, the authors do justify the assumptions made in the paper throughout and in the appendix. Further, it would be great to justify the limitations highlighted above in the weaknesses and questions sections.

**Strengths And Weaknesses:**

**Strengths**

- The problem of task extrapolation is significant and well-motivated. The paper is easy to follow, and the reframing of out-of-support problems as out-of-combination ones is an interesting perspective.
- The framework extends the transductive approach of Netanyahu et al. (2023) from unseen inputs to unseen tasks — a considerably more challenging setting, and one where many existing methods struggle. Length generalization and compositional generalization are important forms of out-of-distribution generalization, and this paper offers a fresh perspective by framing them as relational extrapolation problems.
- The extension to LLMs via relational fine-tuning is novel and represents a promising application of the framework.

**Weaknesses**

- **Strong assumptions made for the general applicability to the three extrapolation settings**: The paper attempts to unify three distinct forms of extrapolation under a single framework, $f_{\theta^{*}}(x) = \Psi(x, f_{\theta_{\text{anc}}}, \phi))$. While the effort to have a unifying framework is appreciated, it is unclear from the provided examples and experiments how broadly the framework applies in each setting. More specifically:
     - **Parameter extrapolation:** The shift $\phi = \Delta\theta = \theta_{\text{target}} - \theta_{\text{anc}}$ is defined as a simple difference in parameters, which seems restrictive. It is not obvious this holds beyond linear or near-linear parameter spaces.
    - **Length extrapolation:** The framework assumes longer tasks can be decomposed into shorter ones, which may hold for linear functions, but is less clear in general. For instance, in the case of 4-digit multiplication where training data only covers 3-digit multiplication, what would $\phi(\theta_{\text{anc}} \rightarrow \theta^*)$ concretely look like?
    - **Compositional extrapolation:** The examples in Section 2.2.3 and experiments in Figure 6 primarily involve compositions of two functions. If training data includes $f_1 \circ f_2$ and $f_3 \circ f_4$, but the test task requires $f_1 \circ f_3 \circ f_2 \circ f_4$, how would the transformation operator $\Psi$ and $\phi$ be defined?

- **Mostly synthetic experiments and simple benchmarks considered:** In line with the above weakness, most of the experiments are done either on very simple synthetic tasks or highly simplified cases of length and compositional generalization benchmarks. This raises questions about the framework's general applicability and scalability to standard benchmarks for each case.

- **Identifiability of underlying structural rules:** In many cases, extrapolation to unseen tasks is not possible as the distribution of training tasks is not sufficient to learn the underlying ground-truth structural rules, causing an identification problem. There is no discussion on when the proposed approach would fail. It would be great to discuss the conditions that the distribution of training tasks must satisfy to enable the successful identification and learning of unseen tasks using the proposed approach, as well as its limitations.

---

> ### Author Rebuttal · Authors · 2026-03-28
>
> # Thank you for your thorough and detailed review
>
> We sincerely thank you for your highly constructive review. The points you raise are incisive, detailed, and will strengthen the paper considerably. We also appreciate your acknowledgment of the strengths of the paper.
>
> ## 1. Multi-Step Composition.
>
> While our experiments focus on single-step extrapolation, **RTE naturally scales to multi-step cases** (e.g., $f(g(h(x)))$) through repeated application of the relational operator: $\Psi(\Psi(x, f_{anc}, g_{trans}), h_{trans})$. The limitation is the combinatorial explosion during decomposition. We will include a discussion paragraph detailing this mechanism and noting how repeated application of the decomposer maintains tractable inference for deeper compositions.
>
> ### Specific Connections:
>
> - Ramesh et al. show that LLMs struggle to generalize to unseen compositions implicitly, but succeed when forced to perform explicit, step-by-step evaluation. RTE mathematically enforces this explicit evaluation without relying on a text-based scratchpad. Their method is similar in spirit to ours; however, we have a fundamentally different method of decomposing the task.
>
> - Zhou et al. show that transformers struggle with length generalization unless the target task aligns with a RASP program. RTE enforces this alignment. In our experiments, the transformation parameter $\phi$ acts as the instruction. By retrieving the prior valid state as $f_{anc}$ and applying the single-step operator $\Psi_{\omega}$, RTE effectively executes recurrent RASP-like programs.
>
> ## 2. Theoretical Guarantees and Nonlinearity
>
> *1. Guarantees*
>
> You are completely correct that **theoretical guarantees require structural assumptions** which we detail in Appendix C. However, as we will discuss, **RTE still outperforms standard learning on task spaces that don't follow the theoretical requirements.**
>
> *2. Nonlinearity*
>
> The framework handles nonlinear settings in two fundamental ways:
>
> First, **the relational operator $\Psi_{\omega}$ is a highly nonlinear neural network.** Even if the parameterization of the shift ($\phi$) is a simple vector, the application of that shift, $\Psi_{\omega}(x,f_{anc},\phi)$, can map to highly complex, nonlinear changes in the functional output space.
>
> Second, we do not compute $\phi$ in the raw parameter space. Instead, we map tasks into a learned latent space (such as the Task2Vec embeddings) where relationships and shifts can be captured more effectively. An example of both mechanisms is the exponential function: for the function $a^x$, a change in $\Delta a$ acts in a nonlinear fashion on the behavior of the function. Yet, RTE still outperforms the inductive models in this case (see Table 1). Furthermore, RTE outperforms even the transductive oracle which has access to the true task parameter because **the latent Task2Vec space maps the function in such a fashion that the resulting change in parameter corresponds to a linear effect on the function's behavior**.
>
> ## 3. Notational Clarity
>
> **$\Psi$ is a fully parameterized neural network** (e.g., an MLP or an LLM) in all instances. Formally, $\Psi_{\omega}(x, f_{anc}, \phi)$ acts as a simulator parameterized by learnable weights $\omega$. Here, $x$ is the $x$-value of the point you wish to predict, $f_{anc}$ is a representation of the anchor function (i.e. either an embedding or a set of points), and $\phi$ is a parameterization of the transformation. In the LLM setting, all this information is contained in a single prompt. We will update the notation throughout the paper to $\Psi_{\omega}$ to make it clear that this is the learned neural model executing the transformation.
>
> ## 4. Can $\Psi$ and $\phi$ be coupled?
>
> Decoupling $\Psi$ and $\phi$ is fundamental. $\phi$ is the descriptor of the change (e.g., "add $+5$ to gravity"). $\Psi_{\omega}$ is the executor that understands how to apply any given $\phi$ to an anchor. By keeping them separate, we can search at test time for the right instructional parameter ($\phi$) and feed it to the generalized executor ($\Psi_{\omega}$) to simulate the unseen task.
>
> To answer the other piece of your question, in the compositional setting $\phi$, the transformation's parameterization, is the representation of another function, analogous to how we parameterize $f_{anc}$.
>
>
> ## Identifiability
>
> You correctly identify the hardest challenge in task extrapolation: identifiability. If we were to learn both the transformation mechanism ($\Psi_{\omega}$) and the task relationships ($\phi$) jointly from scratch, the problem is ill-posed (as discussed in Appendix B.1). **Our approach avoids this failure mode precisely because we assume the structural rules are identifiable during training** (e.g., by utilizing ground-truth relational metadata in the training library or by assuming that the relationships correspond to parameter shifts). We will expand the "Limitations" in the main text to highlight this failure mode.

---

> > ### Author Rebuttal · Reviewer_uWJE · 2026-04-02
> >
> > Thank you for the detailed response; the explanations provided are quite helpful and answer my questions to some extent.
> >
> > I have two main pending concerns, for which I'd like to keep my score.
> > - While the relationship with Ramesh et al. and Zhou et al. makes intuitive sense, it is still not explicit how exactly RTE achieves that, primarily due to the simplicity of examples and experiments used for compositional and length generalization tasks. I'd recommend explicitly demonstrating the usefulness of the RTE framework on the tasks considered in these papers to establish stronger relevance to existing work on task generalization.
> > - The framework assumes significantly more domain knowledge (e.g., ground-truth relationships among training tasks), as well as restrictive assumptions on how tasks are related to each other, making the applicability to real-world generalization tasks more restrictive (as also noted by other reviewers). Additional analysis of how sensitive the proposed framework is to these assumptions would be more useful for applying these ideas in practice.

---

> > > ### Author Response · Authors · 2026-04-07
> > >
> > > # Thank You for a Thorough Review.
> > >
> > > We'd like to thank you for your response; it's very constructive. Your concerns are well taken and align closely with issues we have considered in developing this work.
> > >
> > > We would like to clarify the motivation behind our approach.
> > >
> > > Parameter extrapolation was the problem we intended to solve, motivated by the Vafa et al. papers. In many real-world settings, practitioners encounter out-of-support (OOS) latent task parameters, yet often assume that models will automatically generalize to these settings. A goal of this paper is to challenge this assumption and propose a mechanism for addressing it. **To achieve generalization in OOS settings, it appears essential to explicitly train models to generalize.** As we demonstrate, RTE does this effectively without any relationship labels in the training set.
> > >
> > > While studying parameter extrapolation, we observed strong parallels with **length and compositional generalization** in LLMs. In these domains, generalization is often expected to arise through scaling. However, scaling primarily expands training support and becomes increasingly limited in data-scarce regimes. We believe that leveraging the strategy that worked in parameter extrapolation, i.e. exploiting **latent relational structure** as in RTE, offers a principled path toward enabling generalization in these settings.
> > >
> > > ## Preliminary Extensions
> > > Below, we summarize preliminary results that suggest the challenges you reference are tractable.
> > >
> > > ### (1) Multi-step RTE
> > > While the short rebuttal period prevented us from evaluating RTE on the specific tasks you mentioned, we implemented a multi-step extension in our synthetic functional setting. The setup is the same as Section 3.3 in our paper, where our training functions are compositions of the same primitives as the test functions, but in different combinations, except that the tasks generated are either depth 1, 2, or 3 compositions. This only requires a few changes to RTE.
> > >
> > > In this extension, instead of predicting all pieces simultaneously:
> > > - The Decomposer is autoregressive, implemented using a GRU, and sequentially emits latent task embeddings $(z_1, z_2, z_3)$ by conditioning on the observed target context.
> > > - The Composer remains a one-step functional simulator: $state_{t+1} = \Psi(state_t, \phi_t)$
> > >
> > > **Preliminary results:**
> > > - Inductive baseline (mean MSE): $0.4334 \pm 0.029$
> > > - Recursive RTE (mean MSE): $0.3216 \pm 0.028$
> > >
> > > These results suggest RTE can capture the **recursive structure** of tasks similar to those studied in prior work (e.g., Zhou et al., Ramesh et al.).
> > >
> > > ### (2) Reducing Structural Assumptions
> > > We also explored reducing the structural assumptions required by RTE. A particularly promising setup was evaluated on the CodeIO benchmark.
> > >
> > > When the model is given access to **primitive transformations (but no relationship labels)**, it is able to recover much of the performance. We further implemented an EM-inspired training procedure:
> > >
> > > - Warmup: The model is briefly trained on the primitive transformations.
> > >
> > > - E-Step (Self-Labeling): For composite training tasks, we do not provide the decomposition. Instead, the model scores candidate decompositions using the decomposition which makes selecting the correct answer have the lowest loss. We assign a pseudo-label to the task based on the lowest-loss decomposition.
> > >
> > > - M-Step (Relational Fine-Tuning): The model is updated using these self-generated pseudo-labels, reinforcing its ability to act as a relational simulator.
> > >
> > > Although this procedure incurs a ~4× training slowdown (on a V100), RTE still demonstrates superior performance to the baselines:
> > >
> > > - Standard Few-Shot Baseline: 29.7 \%
> > > - CoT + Majority Voting (N=16): 34.3\%
> > > - **EM RTE: 48.3 \%**
> > > - RTE (full supervision): 57.3 \%
> > >
> > > We found similar results when not giving the model primitives, but instead, a small subset of the relationship labels. With 1/4 of the labels, the model could reconstruct the remaining training labels fairly accurately, resulting in a final accuracy of $52.7\%$.
> > >
> > > One approach to fully removing structural assumptions is to search over the set of relationship labels and select those that minimize error on the sparse test set. This is computationally infeasible but theoretically tractable.
> > >
> > > We also find that RTE naturally supports detection of incorrect structural assumptions and uncertainty quantification. In particular, the availability of sparse in-distribution test data makes conformal inference a natural and effective tool in this setting.
> > >
> > > Although these results are preliminary, they suggest that reducing the structural assumptions required by RTE is feasible. That said, the primary goal of this paper remains to demonstrate that this **OOC perspective can be successfully extended to the important problem of task extrapolation**.
> > >
> > > We thank you again for taking the time to review our work.

---

### Official Review · Reviewer_RuVx · 2026-03-12

**Soundness:** 4
**Presentation:** 3
**Significance:** 3
**Originality:** 3
**Overall Recommendation:** 4
**Confidence:** 4

**Summary:**

This paper presents Relational Task Extrapolation (RTE), an algorithm to enable extrapolation of known tasks to novel tasks. The key idea is to decompose new task into an anchor task and a known transformation. Then a pre-trained relational operator is used to map the anchor-transformation pair to the target task prediction.

Three types of extrapolation are discussed (parameter, length/recursive and compositional). The authors also show the application of RTE in sequence prediction. Extensive experiments are conducted on synthesized tasks as well as LLM in context learning, demonstrating that enhanced task extrapolation ability of RTE compared to inductive learning baselines.

**Compliance With Llm Reviewing Policy:**

Affirmed.

**Final Justification:**

I have read the reviewers comments. I acknowledge that there is theoretical justification based on bound Theorem 1 in the previous work (Netanyahuetal. 2023).  I have increased my score on Soundness and remain positive for this work.

**Key Questions For Authors:**

See weakness

**Limitations:**

Yes

**Strengths And Weaknesses:**

Strength
- It explores an under explored problem of functional extrapolation to unseen regions of task space. The proposed solution makes use of the relational structure between tasks, which is an important observation.
- Significant performance improvement compared to the baselines on both synthesized data and natural language reasoning tasks.
- The example on LLM shows potential of task extrapolation in in-context learning.

Weakness
- The examples given (e.g. extrapolating to higher order polynomial, logical reasoning tasks) are somewhat restricted (clearly belong to the same family), while the claim in the introduction appears bigger. In such a restricted case, the type of transformation is within a finite set (except for parameter extrapolation) that must be observed in the training data. This makes it easier to learn the proxy operator and the decomposer network.   However, in more realistic, open-world cases, such transformation can not be enumerated or described in words.
- Writing is a bit hard to follow. The author should give more practical examples of task extrapolation to demonstrate the real-world application of RTE.
- In compositional extrapolation, the experiments seem to focus on binary composition. How it can extend to higher order needs to be discussed.
- The LLM compositional prompting experiments could use more competitive baselines for prompt-based extrapolation.

---

> ### Author Rebuttal · Authors · 2026-03-28
>
> # Thank you for your Constructive Review
>
> Thank you for your review. It was well thought out, and we appreciated your acknowledgment of the work's strengths. We address your specific concerns below.
>
> ## 1. Scope of RTE
> You raise a fair point regarding the scope of our examples. To start with, parameter extrapolation, which does not suffer the issue of the finite set of available transformations, is likely the most common instance of extrapolation in classic ML. Examples include predicting planetary motion for planets of mass much larger than those seen in training (Vafa et. al.), forecasting a building's structural integrity under extreme temperatures beyond historical data as the world heats up, and sim-to-real transfer in robotics (e.g., teaching a robot to cut vegetables and then quickly adapting it to cut harder materials). **Parameter extrapolation is the core motivation for our work.**
>
> Furthermore, length and compositional extrapolation are ubiquitous in modern LLM reasoning research (see *references*). We agree that in open-world settings, transformations can be vague or difficult to explicitly enumerate in words. RTE does indeed rely implicitly on the idea that the transformation can be parameterized in a meaningful way. However, a core premise of our work is that **we can make these open-world problems tractable by grounding these latent transformations into quantitative proxies** (such as the Task2Vec embeddings demonstrated in our latent experiments). RTE reduces the problem of task extrapolation to meaningfully parameterizing the transformation. However, if no quantitative parameterization of a transformation exists, then RTE would not apply. We will clarify this scope and the transition from finite to open-world transformations in the introduction.
>
> ## 2. Weakness of Exposition
> We agree the exposition can be improved. In the revision, we will introduce concrete, real-world grounding examples (such as those mentioned above) directly in Sections 1 and 2 to better contextualize the theory and methods.
>
> ## 3. Higher Order Compositions
> While we did not cover thisthoroughly in the paper, **RTE natively supports higher-order compositions** (e.g., $f(g(h(x)))$) through recursive application of the relational operator: $\Psi(\Psi(x, f_{anc}, g_{trans}), h_{trans})$. The limitation is not the operator's capacity, but the combinatorial explosion during the decomposition phase. We will include a discussion paragraph detailing this mechanism and noting that heuristic mechanisms such as beam search and repeated application of the decomposer network can maintain tractable inference for deeper compositions.
>
> ## 4. Stronger Baselines
> While prompt-based baselines targeting task extrapolation are scarce, we agree a stronger comparison is needed. To address your concern, we evaluated the few-shot baseline using chain-of-thought (CoT) prompting combined with Majority Voting for the CodeIO task in Section 4.2.
>
> As shown below, RTE still demonstrates superior performance:
> - Standard Few-Shot Baseline: 29.7 \%
> - CoT + Majority Voting (N=16): 34.3\%
> - **RTE (Ours): 57.3 \%**
>
> We will include these results in Table 5 and Section 4.2 of the final manuscript.
>
> *References:*
>  - *Zhou, Hattie, et al. "What Algorithms can Transformers Learn? A Study in Length Generalization." The Twelfth International Conference on Learning Representations.*
>  - *Ramesh, Rahul, et al. "Compositional Capabilities of Autoregressive Transformers: A Study on Synthetic, Interpretable Tasks." Forty-first International Conference on Machine Learning.*
>
> *Zhou et. al show that LLMs struggle with length extrapolation, unless the recursive step follows a specific structure, known as "RASP" program. RTE enforces this structure implicitly.*
>
> *Ramesh et. al show that LLMs struggle with compositional extrapolation unless prompted to explicitly write out the steps of the composition one at a time. RTE takes a different approach to make the composition explicit but follows a similar principle.*

---

> > ### Author Rebuttal · Reviewer_RuVx · 2026-04-05
> >
> > Thank you for the detailed response. The authors had given more thorough explanations to the scope of RTE (Q1), discussed recursive composition strategy (Q3) and added a baseline using CoT + majority voting (Q4) to support the performance superiority of the proposed method.
> >
> >  Nevertheless, while the paper introduces new ways of modeling task extrapolation, it is mainly supported by empirical results on well-designed experimental settings, rather than theoretical analysis or open-world benchmark, I will maintain the already positive score.

---

> > > ### Author Response · Authors · 2026-04-07
> > >
> > > # Thank you for engaging with our work
> > >
> > > We are pleased that our responses have answered your questions. Thank you again for your thoughtful feedback.
> > >
> > >  **We would like to note that we did provide one theoretical guarantee in Appendix C.** Under some regularity conditions on the structure of the task space, there exists an error bound on this kind of extrapolation that cannot be achieved by standard inductive approaches. As we mentioned in our initial rebuttal to Reviewer uWJE, RTE still worked empirically better than standard learning in settings where these regularity conditions don't hold.
> > >
> > > Also, we would like to highlight that **RTE is not meant to be a black box solution to all task extrapolation problems.** Successful extrapolation requires the presence of relational structure in the data.  Our goal is to address the subset of task extrapolation problems that have such structure, which RTE converts into an **OOC (out-of-combination) structure**, which includes many practically relevant OOS settings. We believe this represents an important and underexplored class of problems.
> > >
> > > Thank you again for taking the time to review our work.

---

### Decision · Program_Chairs · 2026-04-30

**Decision:**

Accept (regular)

**Comment:**

This paper proposes RTE, a relational approach to task extrapolation that decomposes unseen tasks into anchor tasks and transformations, enabling generalization to out-of-support task distributions.

Reviewers appreciate its
- clear motivation for addressing task extrapolation beyond interpolation regimes,
- novel relational formulation that reframes extrapolation as an out-of-combination problem,

However, concerns focus on
- limited evaluation on simplified or synthetic benchmarks, raising questions about real-world applicability,
- strong structural assumptions (e.g., task decomposition and identifiability),
- insufficient theoretical grounding and unclear generalization to more complex task spaces,
- scalability issues, including search over anchor–transformation pairs and reliance on auxiliary models.

The rebuttal provides additional clarifications and experiments, including stronger baselines, extended discussions on multi-step composition, and analysis of structural assumptions, which address several concerns. As reflected in the final justifications, most reviewers acknowledge that their concerns have been addressed and maintain positive scores; the remaining concern (Reviewer JJTp) pertains to core premise setup, which have been clarified by the authors and are not considered to undermine the overall contribution of the work.

For the final revision, the authors are encouraged to incorporate key rebuttal clarifications into the main paper, including
- clearer discussion of applicability and scope of RTE,
- additional evaluation added during the rebuttal,
- explicit analysis of structural assumptions and identifiability,
- improved exposition and discussion of limitations.